biomechanics/civil engineering

echinoids, biomechanics, shell structures, bioinspiration, structural optimization

**Author for correspondence:**
Valentina Perricone
e-mail: valentina.perricone@unicampania.it

# Flexible sutures reduce bending moments in shells: from the echinoid test to tessellated shell structures

Francesco Marmo[1], Valentina Perricone[3], Arsenio Cutolo[1,2], Maria Daniela Candia Carnevali[5], Carla Langella[4] and Luciano Rosati[1]

[1]Department of Structures for Engineering and Architecture, and [2]LIMITS Laboratory, University of Naples Federico II, Napoli, Italy
[3]Department of Engineering, and [4]Department of Architecture and Industrial Design, University of Campania Luigi Vanvitelli, Aversa, Italy
[5]Department of Environmental Science and Policy, University of Milano, Milan, Italy

VP, 0000-0002-1410-2695

In the field of structural engineering, lightweight and resistant shell structures can be designed by efficiently integrating and optimizing form, structure and function to achieve the capability to sustain a variety of loading conditions with a reduced use of resources. Interestingly, a limitless variety of high-performance shell structures can be found in nature. Their study can lead to the acquisition of new functional solutions that can be employed to design innovative bioinspired constructions. In this framework, the present study aimed to illustrate the main results obtained in the mechanical analysis of the echinoid test in the common sea urchin *Paracentrotus lividus* (Lamarck, 1816) and to employ its principles to design lightweight shell structures. For this purpose, visual survey, photogrammetry, three-dimensional modelling, three-point bending tests and finite-element modelling were used to interpret the mechanical behaviour of the tessellated structure that characterize the echinoid test. The results achieved demonstrated that this structural topology, consisting of rigid plates joined by flexible sutures, allows for a significant reduction of bending moments. This strategy was generalized and applied to design both free-form and form-found shell structures for architecture exhibiting improved structural efficiency.

# 1. Introduction

In structural engineering, the use of efficient structural systems provides the double advantage of reducing the impact of the construction processes on the environment and obtaining structures that resist extreme loadings (e.g. earthquakes, volcanic eruptions and floods). Owing to their geometric stiffness and reduced mass, shell structures represent one of the most efficient structural systems to resist actions that generally affect constructions. The efficiency of shell structures is mainly owing to their geometry since this is characterized by a thick curved surface having two large dimensions (mid-surface) and a smaller one (thickness). However, similarly to several structural systems, their efficiency is also related to the distribution of internal forces, namely membrane forces, bending moments and out-of-plane shears.

In this respect, structural optimization techniques, widely employed to design shell structures and referred to as form-finding methods, aim at defining the geometry of the shell mid-surface so that applied loads can be equilibrated mainly by membrane forces [1]. In fact, structural solutions that minimize bending moments in favour of membrane forces are desirable when optimization of material exploitation is a primary goal. However, the optimal configuration of structures is strongly related to every specific load case and is unlikely adaptable to all load conditions that a real structure is expected to resist.

On the other hand, a large amount of optimized shell structures, designed to effectively sustain a variety of loading conditions, can be found in nature. Numerous living organisms employ protective shells against predators or external abiotic loadings. Some of them, such as *Acanthocardia*, *Pecten* and *Tridacna* shells, use corrugation as a strategy to optimize structural performance [2]. Actually, the efficiency of shell corrugation has been successfully employed in the construction industry to increase shell geometric stiffness and strength against loading variability with a limited increase of structural thickness and weight [3].

A completely different solution is shell tessellation that amounts to partitioning the shell mid-surface into a series of patches united by compliant joints. Shell tessellation does exist in numerous organisms and provides a range of interesting mechanical properties at different scales (from molecular to macroscopic arrangements), such as crack propagation prevention, flexibility, and protection for biological shells and armours [4]. An outstanding example of a tessellated shell is observed in echinoids, known as sea urchins, that employ a coherent and resistant dome-shaped and tessellated shell structure (figure 1), also defined as test or *echinodome* [5–8]. It is composed of a series of skeletal plates joined by skeletal protrusions and collagenous sutures [8–13]. Each plate consists of high-magnesium calcite material arranged in a porous three-dimensional lattice-like meshwork (stereom) [14–16]. This design fulfils several mechanical and biological functional principles acting as a resistant, lightweight and load-bearing system adapted to withstand biotic (e.g. predatory attacks) and abiotic (e.g. environmental forces such as fluid flow and pressure) mechanical stresses [2,17–22]. Additionally, the tessellated configuration is functional for the echinoid test growth since sutures guarantee a space between plate margins (plate gapping) so that each plate can continuously expand while interacting with the adjacent ones [23].

The structural behaviour of the regular echinoid test has drawn the attention of researchers. Previous literature investigated the mechanics of echinoid test by focusing on the structural strengthening provided by suture ligaments [10] without considering tessellation and flexible sutures [17]. Understanding the role of tessellation and flexible sutures in increasing the strength of echinoid tests and shell structures is the main aim of the present study.

The experiments carried out by Ellers *et al.* [10] revealed that the echinoid test is strengthened by flexible collagenous ligaments. Philippi & Nachtigall [17] conducted a pioneering finite-element analysis (FEA) describing the behaviour of the regular echinoid test (*Echinus esculentus*) under different loads. Their studies highlighted the structural load-bearing efficiency of the echinoid test and interpreted its peculiar spherical shape as the most adapted form to sustain tensile stresses resulting from tube-feet activity. Nonetheless, they modelled the echinoid test as a monolithic structure neglecting its tessellated configuration and suture flexibility.

In addition, the structural and functional adaptations of these animals represent useful solutions that can be abstracted and transferred to the design of new technical structures by means of an interdisciplinary approach known as 'Biomimetic' [24–29]. Recently, the echinoid design has revealed a high potential in transferring biomimetic solutions, notably in building constructions [13,30–32]. The ICD-ITKE of the University of Stuttgart successfully demonstrated the application of different echinoid structural principles (such as plate arrangement, growth process and material differentiation) [12] in a series of demonstrative temporary pavilions [33–37] and permanent buildings [38–41].

In this framework, the present research pursues the biomechanical analysis of the global behaviour of the test in the regular sea urchin *Paracentrotus lividus* (Lamarck, 1816) with the main purpose of

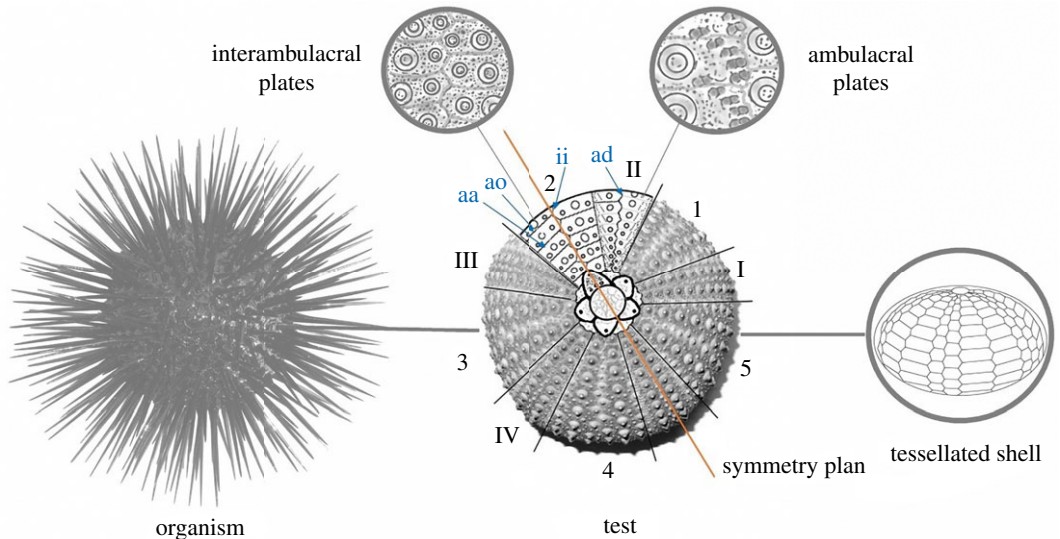

**Figure 1.** Regular echinoid. Schematic reconstruction of a regular echinoid, its skeletal test in aboral view and functional characteristic of a tessellated shell structure. The aboral view shows the subdivision in ambulacral (I–V) and interambulacral (1–5) zones along with respective plates and the symmetry plan. aa = adapical suture; ao = adoral sutures; ad = adradial suture: ii = interradial suture.

identifying new functional strategies that can be abstracted and transferred to the design of new shell structures for building constructions. Specifically, we hypothesize that the structural organization of the echinoid test, which is a composition of multiple rigid plates joined by flexible sutures, can significantly reduce its bending moments resisting external loads mainly by membrane forces, without compromising the global deformability and stability of the shell. To confirm this hypothesis, we employed a series of experimental tests and FEAs focusing on the role played by tessellation on reducing the magnitude of bending moments in the shell.

Employing basic mechanics and visual survey results, we inferred that the tessellated organization of the echinoid test avoids rigid-body mechanisms, i.e. flexible sutures do not form articulations and inhibit rigid movements. Successively, a series of experimental results conducted on single plates and on plate–plate pairs were used to verify the rotational behaviour of sutures. A photogrammetric survey of the *P. lividus* test provided a parametric geometric model, used to numerically analyse the mechanical behaviour of the echinoid test. By comparison with results pertaining to a monolithic shell, i.e. having the same geometry and mechanical properties but rigid sutures, FEAs showed that flexible sutures reduce bending moments. The structural features that were identified as responsible for this peculiar structural behaviour were abstracted and applied to design both free-form and form-found shell structures. Although these structures are characterized by a completely different scale, material and shape, besides being subjected to completely different loading conditions with respect to the echinoid test, a series of numerical models showed that bending moments are significantly reduced by the presence of flexible joints independently from the specific shell structure features.

This article is organized as follows: after recalling that membrane forces and small bending moments mainly characterize the internal stresses of optimized shell structures, §3 is dedicated to the analysis of the *P. lividus* test shell. Section 2.1 is focused on its geometry and how it influences the kinematical behaviour of the shell; §2.2 presents the results of the three-point bending tests carried out on plates and sutures; §2.3 describes the photogrammetric survey of the test, adopted as a geometric model for the FEAs reported in §2.4. Section 3 shows how the structural behaviour of the echinoid test can be transferred to large-scale shell structures, both form-found (§3.1) and free-form (§3.2). Finally, conclusions are drawn in §4.

## 2. Investigation of the echinoid test

The echinoid test can be modelled and described as a tessellated shell structure [7,8]. Shell structures are geometrically defined as three-dimensional-curved solids in which thickness is much smaller than the other two dimensions. For this reason, they are schematized by a curved mid-surface and the corresponding thickness $t$. Such structures are capable to withstand applied loads by *membrane* (or

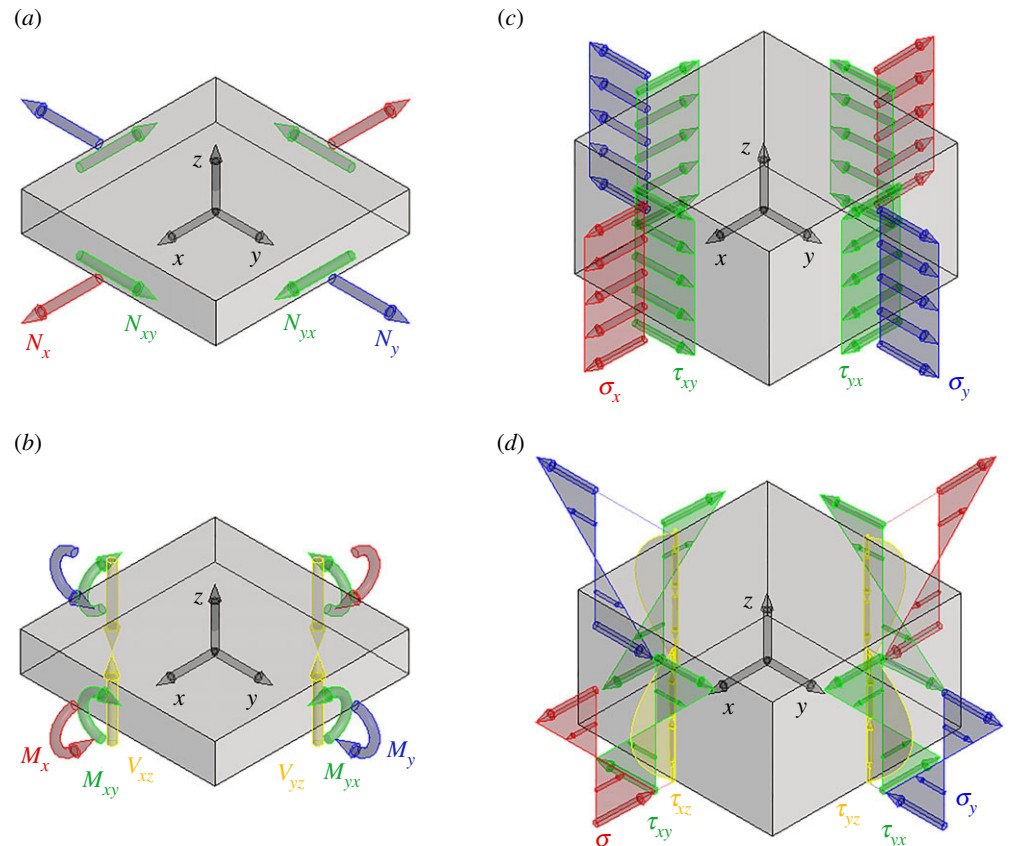

**Figure 2.** Internal force components in shells and the corresponding Cauchy stresses: (*a*) membrane forces, (*b*) bending moments and out-of-plane shear forces, (*c*) Cauchy stresses associated with membrane forces, and (*d*) Cauchy stresses associated with bending moments and out-of-plane shear forces.

*tangential*) *forces*, *bending moments* and *out-of-plane shear forces* [42,43]. The membrane forces and bending moments are described by means of generalized stress tensors:

$$\mathbf{N} = \begin{bmatrix} N_x & N_{xy} \\ N_{yx} & N_y \end{bmatrix} \quad \text{and} \quad \mathbf{M} = \begin{bmatrix} M_x & M_{xy} \\ M_{yx} & M_y \end{bmatrix}, \tag{2.1}$$

while the out-of-plane shears have only the two components $V_x$ and $V_y$, e.g. figure 2*a,b*. Here subscripts refer to the force and moment components represented in a three-dimensional reference frame having axes $x$ and $y$ tangential to the shell mid-surface and directed along the principal direction of the shell curvature, and axis $z$ orthogonal to the shell mid-surface.

Membrane and out-of-plane shear forces, as well as bending moments, are defined as the resultant forces and moments of the Cauchy's stresses acting on the faces of an infinitesimal portion of the shell mid-surface, whose depth is the shell thickness. Specifically, Cauchy's stresses in shells are [43]:

$$\sigma_x = \left( \frac{N_x}{t} + \frac{12M_x}{t^3}z \right) \frac{1}{1 - z/r_y}, \tag{2.2a}$$

$$\sigma_y = \left( \frac{N_y}{t} + \frac{12M_y}{t^3}z \right) \frac{1}{1 - z/r_x}, \tag{2.2b}$$

$$\sigma_z = 0, \tag{2.2c}$$

$$\tau_{xy} = \tau_{yx} = \frac{N_{xy} + N_{yx}}{2t} + \frac{6(M_{xy} + M_{yx})}{t^3}z, \tag{2.2d}$$

$$\tau_{xz} = \frac{3V_x}{2t}\left( 1 - \frac{4z^2}{t^2} \right) \tag{2.2e}$$

and

$$\tau_{yz} = \frac{3V_y}{2t}\left( 1 - \frac{4z^2}{t^2} \right), \tag{2.2f}$$

where $r_x$ and $r_y$ represent the principal radii of the shell curvature.

The membrane and bending tensors $N$ and $M$ become symmetric, i.e. $N_{xy} = N_{yx}$ and $M_{xy} = M_{yx}$ when the shell has a thickness that is much smaller than its principal radii, i.e. $t \ll r_x$ and $t \ll r_y$. In this case, the stress components $\sigma_x$, $\sigma_y$ and $\tau_{xy}$ associated with null bending moments are uniform along $z$, e.g. equations (2.2a)–(2.2d) and figure 2c. In other words, membrane forces are associated with a uniform exploitation of material along the shell thickness. Conversely, bending moments produce a linear variation of the stress components $\sigma_x$, $\sigma_y$ and $\tau_{xy}$ along $z$, with a null mean value, e.g. equations (2.2a)–(2.2d) and figure 2d. Thus, they are associated with uneven exploitation of material along the shell thickness: when shells are subjected to high bending, points near the shell mid-surface experience small stress values and do not significantly contribute to the load-bearing capacity of the structure. Finally, equations (2.2e) and (2.2f) describe how out-of-plane shear forces are related to the Cauchy stress components $\tau_{xz}$ and $\tau_{yz}$, these last ones having a parabolic law of variation along the shell thickness (figure 2d).

Consequently, an optimized shell structure equilibrates external loads mainly by membrane forces, while bending moments are relatively small. This solution is achieved in funicular shells, i.e. shells having their mid-surface aligned with the resultant stress lines. Furthermore, this concept is employed by form-finding procedures to optimize the form of a shell with respect to a given loading condition [1,3,44–46]. However, the optimized structural form is strongly related to the specific loading condition. This prompts the use of different structural typologies that better adapt to the variety of loading conditions that the structure is subjected to during its life.

In this regard, biological structures are structurally and functionally adapted to withstand a variety of loading conditions imposed by biotic and abiotic environmental factors. Although these are not the only factors that drive environmental adaptation, evolution implemented interesting solutions that optimize geometry, structural organization and employment of material in natural shells, providing more efficient and lightweight structures [2,11].

In sea urchins, it is highly plausible that an adaptive strategy to increase both lightness and resistance contributed to the evolution of the skeletal test. Because of the low metabolic rate in sea urchins, the biomineralization energy source is limited and must be exploited in the most efficient way [47]. The possession of a resistant test, and consequently effective visceral protection, increases the chance of echinoids surviving. In this light, it is here demonstrated that the structural efficiency of the echinoid test, which is composed of multiple plates joined by collagenous sutures, can be attributed to its capability to equilibrate external loads mainly by membrane forces and limited bending. As it has been previously described, this structural behaviour optimizes the use of material and the mechanical performances of the shell. Although described and partially motivated by previous research [10,17], presently, to our knowledge, no studies have either explored nor demonstrated in detail the mechanical behaviour of the overall test structure by considering its tessellated configuration with flexible sutures.

## 2.1. Visual survey

To identify the main structural features of the echinoid test and select the ones that significantly contribute to its global structural behaviour, specimens of *P. lividus* were analysed in depth both at the macro- and microscale using a stereomicroscope (Leica M205C) and high resolution-scanning electron microscope (JEOL 6700F 250 MK2).

The samples were completely digested via 0.1 N NaOH treatment (for about 1–2 weeks), which completely removed the organic components and provided perfectly clean disassembled plates. Samples were then washed three times in deionized water to remove caustic remains, air-dried and analysed according to standard scanning electron microscopy (SEM) methods. Different plates were isolated, suitably sectioned and processed for observation in SEM (at 5 or 10 kV).

Macroscopically, the overall echinoid test forms an evident pentaradial structure, where pseudo-hexagonal plates are regularly arranged according to 10 double series of plates, respectively, representing alternating five ambulacral and five interambulacral zones (figure 1) [48,49]. The ambulacral plates are typically pierced by pores (double series) for tube-feet emergence. These pore-pairs are located and aligned along the outer (adambulacral) margins of the plates. Owing to the regular alignment, each ambulacral and interambulacral zone displays longitudinal (radial or meridional) and latitudinal (circumferential) sutures. Radial sutures are: (i) the perradial suture, between the two series of ambulacral plates; (ii) interradial sutures at the midline of each interambulacral zone; and (iii) the adradial suture, i.e. the longitudinal joints between interambulacral and ambulacral plates. Circumferential sutures are adapical, i.e. the upper edge of a plate, and adoral

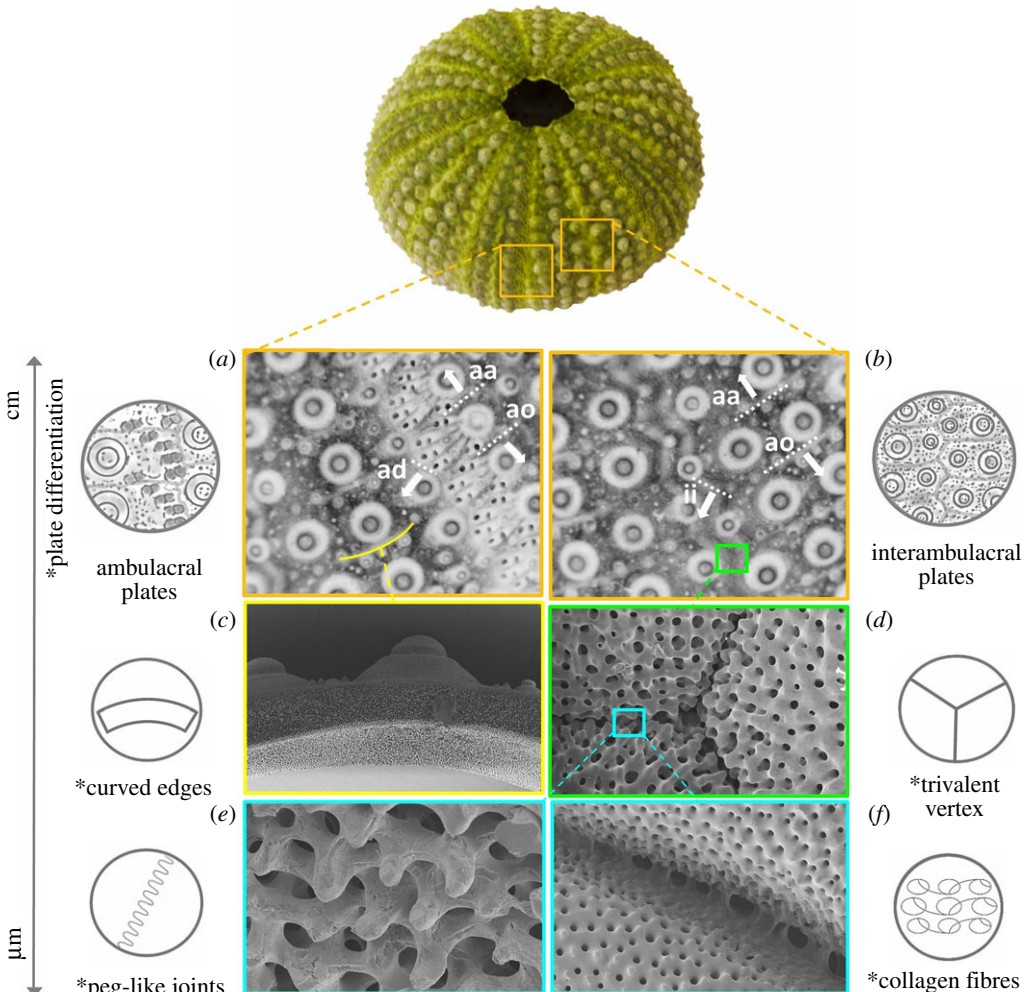

**Figure 3.** *Paracentrorus lividus* test: functional details from macro to microscale. Test with plate diversity in (*a*) ambulacral plates with adapical and adradial sutures and (*b*) interambulacral plates with adradrial and interratial sutures; (*c*) plates with curved edges indicated in (*a*); (*d*) trivalent vertex arrangement magnified from (*b*); (*e*) knob-like protrusions; and (*f*) collagen fibres magnified from (*d*). Asterisk = functional features. aa = adapical suture; ao = adoral sutures; ad = adradial suture: ii = interradial suture.

sutures, the lower one [50] (figure 3*a*,*b*). Since all plates bear movable spines, the outer surface of the plate displays suitably rounded tubercles on which the spines are articulated by ball-and-socket joints [13].

Microscopically, SEM surveys on the *P. lividus* test showed high microstructural variation within the plates. Considering an interambulacral plate (figure 4*a*), the skeletal material tends to increase in density in different regions, such as tubercles and basal zone, and to specialize its microstructure in the zones more subjected to directional forces, i.e. the sutural ones (figure 4*b*). At the sutures, these external bands are characterized by a regular porous arrangement (galleried stereom) [14], which is geometrically ordered and regularly oriented according to the junction direction (figure 4*b*,*e*). The suture area terminates with knob-like trabecular protrusions that allow interlocking between the adjacent plates (figure 4*c*,*d*,*e*). According to the description of Mancosu & Nebelsick [50], the sutural micromorphology among the different radial and circumferential regions is slightly variable; however, these differences can be considered negligible at the global scale. Moreover, plates are connected at sutures by short and strong articular ligaments, consisting of parallel bundles of densely packed collagen fibrils. In regular echinoids, these sutures remain 'open', up to the adult stage; thus, it is highly plausible that they actively influence the mechanical behaviour of the echinoid test as proved in the turtle carapace [51].

Considering the sutural collagen bundles, it is also important to emphasize that, apart from rare exceptions, the vast majority of connective tissues in living echinoids and in all the other echinoderms consist of peculiar collagenous tissues, called mutable collagenous tissues (MCT), characterized by

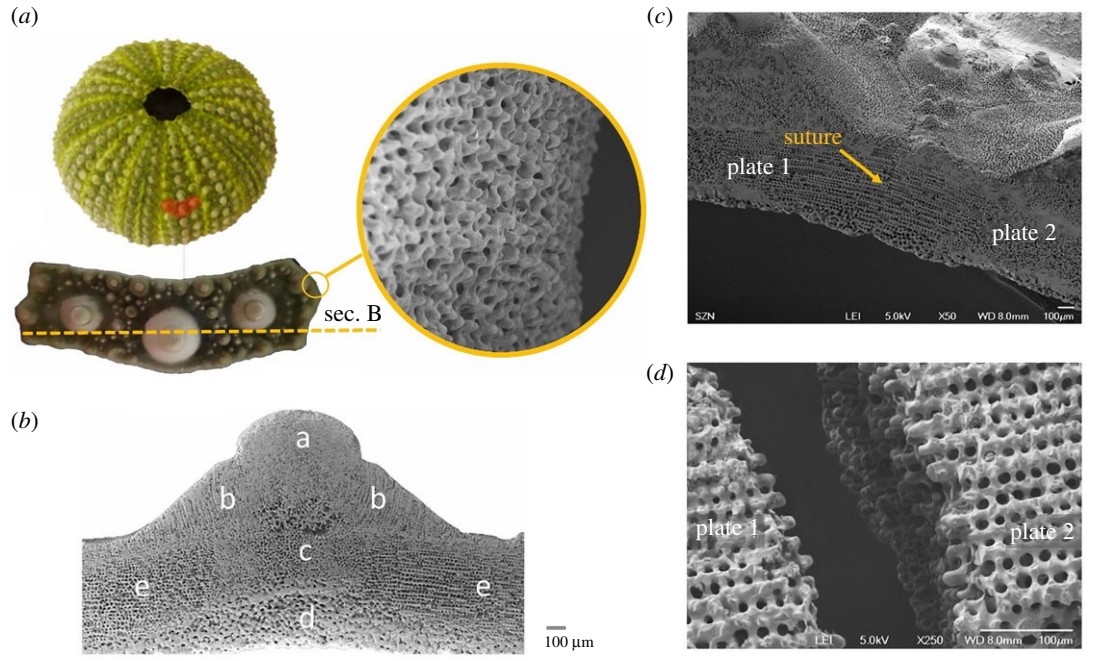

**Figure 4.** Macro and microstructure of an interambulacral plate. (a) Interambulacral plate with detail of knob-like protrusions, extracted from *P. lividus's* test. SEM micrographs showing: (b) plate transversal section with microstructural variability (a, imperforated; b, tubercle galleried; c, labyrinthic; d, perforated; e, suture galleried) identified by a dashed line; suture area of (c) joined and (d) divided plates.

unique mechanical behaviour [52,53]: in fact, they can modulate their passive mechanical properties (tensile strength, stiffness and viscosity) passing from a rigid state to a semi-flexible or to an even very pliant condition according to the functional needs and under direct neural control [52,53]. MCTs are ubiquitous in all extant echinoderms most of all in the form of dermal connective tissue in the body wall, articular ligaments interconnecting skeletal components and tendons linking muscles to skeletal elements [52,53]: in all these cases, MCTs present analogous functional roles as dermis, ligaments and tendons in vertebrates, but their mechanical properties do change in an exceptionally rapid (less than 1 s) and drastic way (up to two orders of magnitude, e.g. in spines) [52]. The massive presence of MCTs is considered a distinctive feature of the phylum Echinodermata, and therefore, although not yet demonstrated, it is very plausible that the collagenous sutural ligaments of *P. lividus* test consist of MCTs.

Visual survey, together with knowledge provided by the literature [8–10,14,15,17–21,30,31,54], have highlighted four major functional features that contribute to the stability of the echinoid structure and influence its mechanical behaviour, namely: knob-like protrusions and collagenous ligaments, trivalent vertex arrangement of joints, curved geometry of plates and edges (figure 2c–f).

Sutures are characterized by the presence of interdigitated articular surfaces (knob-like protrusions) and are bound together by bundles of short collagenous ligaments [9,10]. Owing to their microstructure, knob-like protrusions can prevent shear movements between plates. Collagenous ligaments, on the other hand, avoid plate separation. Hence, the combined effect of knob-like protrusions and collagen fibres make sutures behave as cylindrical hinges, i.e. they allow relative rotation between plates while preventing sliding. These hinges do not compromise the global stability of the echinoid test since plates are arranged following the trivalent vertex principle, in which three plates meet at one point [30,31,54]. As shown by Wester [30], the subdivision of a shell structure in different plates and their arrangement in trivalent vertices (Y-shaped) provides stability to the echinoid test as occurs in a panel structure. Interestingly, Wester describes and compares the dualistic nature of the panel structure with pure lattice structures: as the triangular mesh stabilizes lattice structures, three lines of support are necessary for panel structures to avoid rigid mechanisms of a plate in a three-dimensional space. Although the trivalent vertex pattern is enough to prevent rigid mechanisms in closed plate structures, the echinoid tests are opened at the oral side, with the mouth (peristome), and the aboral side, with the anus (periproct). These two discontinuous areas, in particular the peristome, are remarkably large and not negligible; hence, the trivalent vertex principle is not sufficient to guarantee the stability of the overall shell structure.

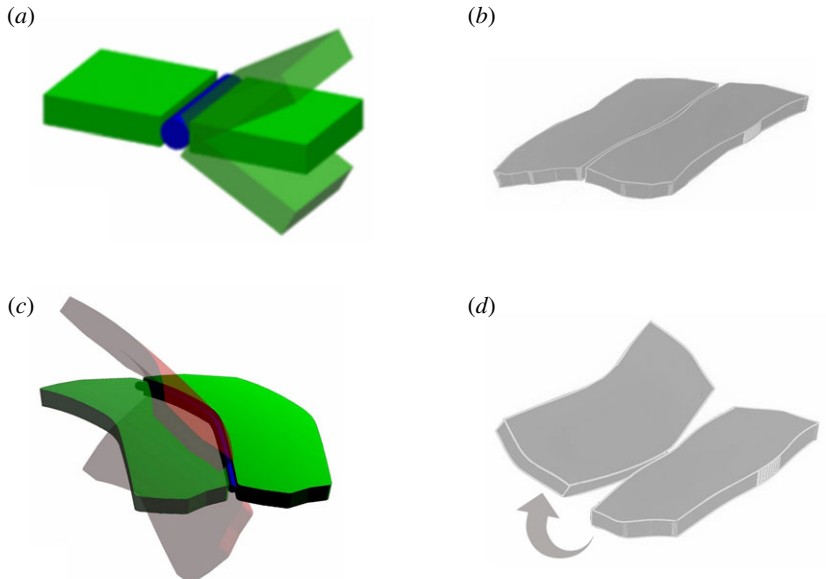

**Figure 5.** Mechanical behaviour of a curved plate. (*a*) Flat plates can undergo relative rotation by keeping adjacent edges in contact. (*b*) Joined echinoid plates with curved geometry. (*c,d*) Relative rotation between curved plates causes separation of adjacent edges.

Different from the structures described by Wester, the joints between plates constituting the echinoid test have a curved geometry. Owing to the curvature of the intersection lines, relative rotation between two adjacent plates is coupled with relative displacements between adjacent edges. As illustrated in figure 5, flat plates joined by sutures can easily undergo relative rotation still fulfilling compatibility between adjacent edges (figure 5*a*); on the other hand, the rotation between two adjacent curved edged plates is always associated with a relative displacement between their edges (figure 5*c,d*). Technically, a series of cylindrical hinges aligned with the curved boundary of the plates avoids the existence of the screw axis for the relative rigid motion of plates [55]. Hence, if the separation between adjacent edges is hampered, for example, by the ligaments present between echinoid plates, the global mechanisms and rotations are also avoided when the trivalent vertex principle cannot be applied: curved hinges avoid rotation mechanisms even with only one line of support per plate, still avoiding the transfer of bending moments.

## 2.2. Three-point bending test

To characterize the mechanical properties of plates and sutures, a series of three-point bending tests were carried out both on single plates and on plate–plate pairs joined by a suture. Five specimens of *P. lividus* collected in the Gulf of Naples and kept for one week in seawater at 18–20°C were cut into small portions of ambitus interambulacral plates, obtaining, from each specimen, two samples of single plates and two samples of plate–plate pairs joined by an interradial suture. Successively, by means of a customized cutting system, these samples were cut in the shape of a beam with a rectangular cross-section (size $b$(base) $= 1.9 \div 3.7$ mm by $h$(depth) $= 0.9 \div 1.2$ mm). Plate–plate pairs were cut so that the suture was positioned in the middle and orthogonal to the beam's longitudinal axis. The locking effect on rotations produced by the suture curvature was negligible owing to the small width of the plate–plate samples (see §2.1). Each sample was tested with the aim of a TA Instruments ElectroForce 200 N - 4 motor Planar Biaxial Test Bench. Ad-hoc supports have been manufactured in thermoplastic material (ABS) via a three-dimensional printing system (Stratasys Object 30 Pro) to conduct three-point bending tests over a span of 5.0 mm, e.g. figure 6. Samples were loaded at midspan along the weaker axis, orthogonal to the shell mid-surface, at a crosshead speed of 0.1 mm s$^{-1}$ while keeping the samples wet to preserve the mechanical properties of collagenous ligaments.

In order to verify if the collagenous sutures of the echinoid test consisted of MCTs and were therefore subjected to dynamic *mutability* phenomena, preliminary tests with elevated K$^{+}$ concentrations (100 mM K$^{+}$ in seawater) were carried out; this treatment has been usually employed in biomechanical tests to neurally induce stiffening in MCTs [52,53,56]. For this reason, the sutural collagenous ligaments were

(a) 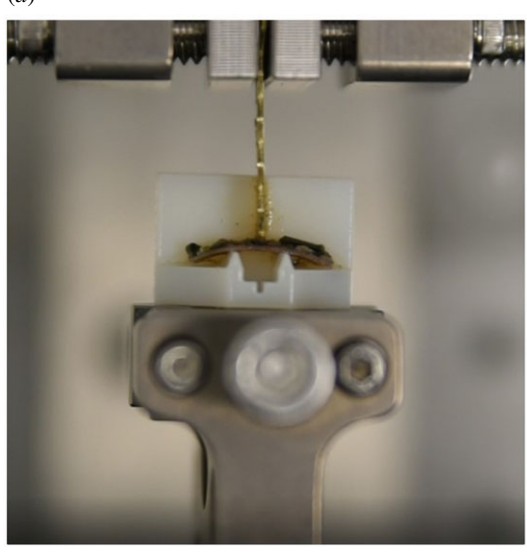  (b) 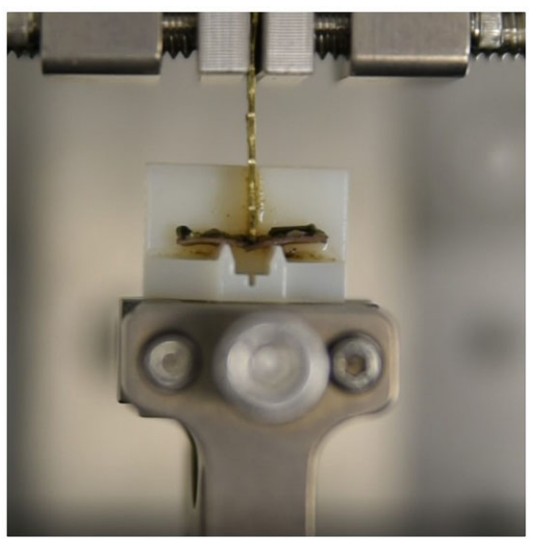

**Figure 6.** Experimental set-up. Three-point bending test executed by a TA Instruments ElectroForce 200 N - 4 motor Planar Biaxial Test Bench on ad-hoc supports manufactured in thermoplastic material (ABS) via a three-dimensional printing system (Stratasys Object 30 Pro). Plate–plate sample before (a) and after (b) testing.

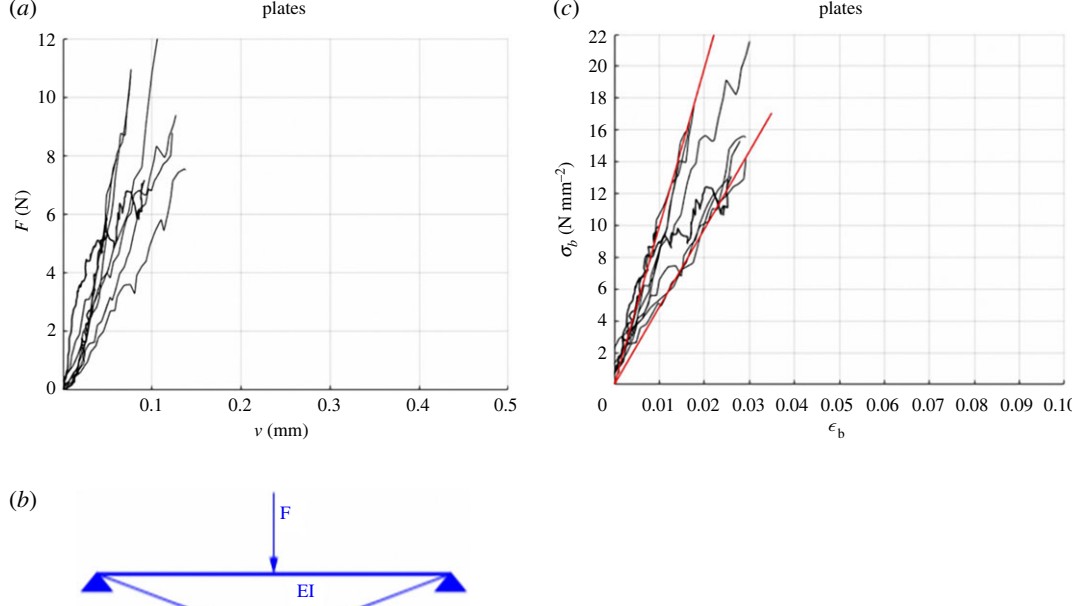

**Figure 7.** Experimental curves for single plates: force–displacement curves (a) are interpreted by the scheme (b) to obtain the stress–strain curves (c).

tested in two experimental conditions: (i) normal filtered seawater; and (ii) seawater with 100 mM $K^+$, which was expected to modify the mechanical state of MTCs [52].

The results of the three-point bending tests are shown in figures 7 and 8. In particular, figures 7a and 8a illustrate the load–displacement (F–v) curves for a single plate and for plate–plate pairs, respectively. Single plates were characterized by a linear elastic response, with brittle failure. On the contrary, plate–plate pairs were very compliant for small values of rotation $\phi$ and became stiffer until a peak value of force was reached. For higher displacements, a softening behaviour was observed showing the high displacement capacity of the plate–plate system. No relative sliding between plates was observed, denoting that failure was governed by bending rather than shear.

Collapse of single plates was attained at $F = 7.5 \div 12$ N and $v = 0.076 \div 0.138$ mm. The flexural strength and the corresponding apparent Young modulus of plates, imagined as homogeneous

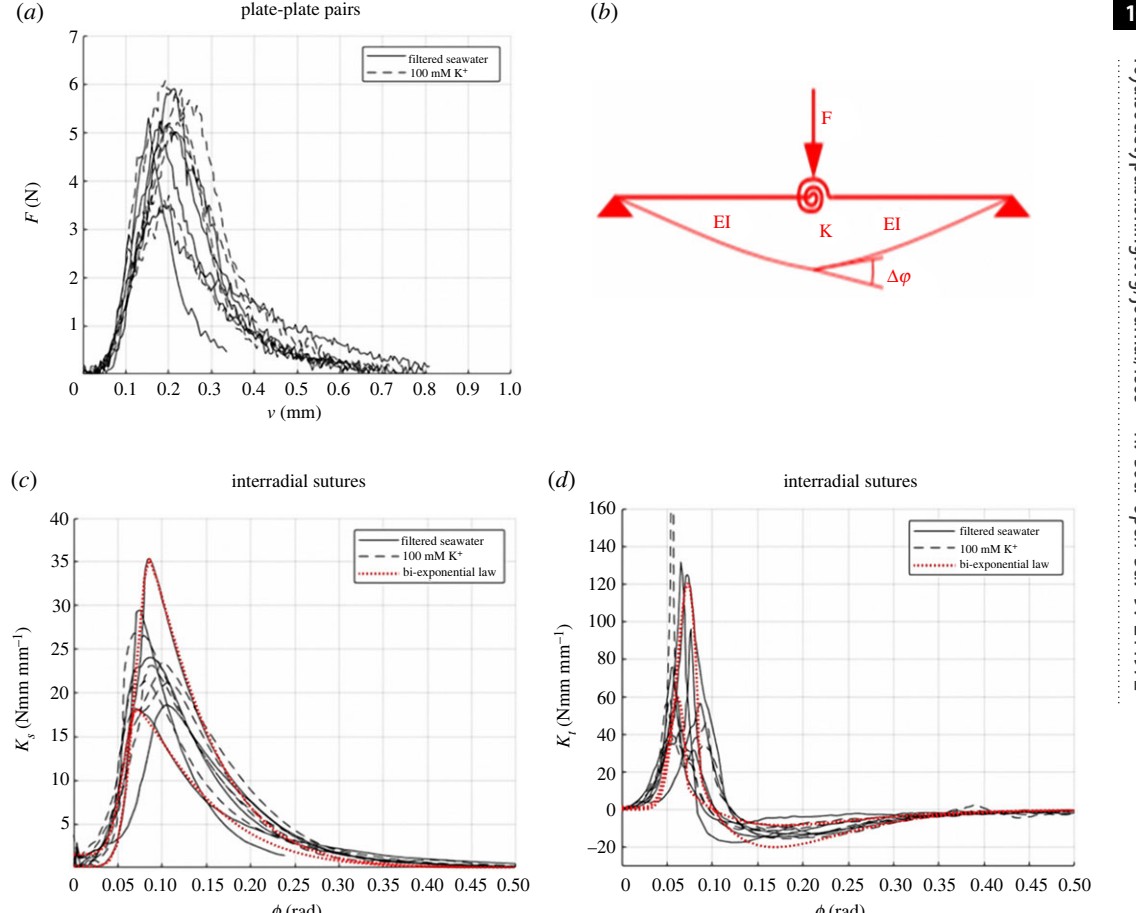

**Figure 8.** Experimental curves for plate–plate pairs: force–displacements curves (*a*) are interpreted by the scheme (*b*) to obtain the secant stiffness–relative rotation (*c*) and the tangent stiffness–relative rotation (*d*) curves. Solid curves refer to plate–plate pairs in natural states while dashed curves refer to plate–plate pairs treated with K$^+$ solution. Red dotted curves are relevant to a parametric description of the two curves by the analytical models for the secant stiffness (equation (1.3)) and the tangential stiffness (equation (1.4)).

and isotropic, were estimated by considering the static scheme of figure 7*b* and accounting for the actual size of the cross-section of each sample. According to this model, the stress–strain law of the extreme points of the midspan cross-section was obtained by evaluating $\sigma_b = 3F\ell/2bh^2$ so that, being $v = F\ell^3/48EI$ with $I = b \cdot h^3/12$, resulting in $E = F\ell^3/4bh^3v$ and $\varepsilon_b = \sigma_b/E = 6vh/\ell^2$. The corresponding stress–strain response of plates is reported in figure 7*c* and the estimate of the apparent Young modulus is $E = 735 \pm 255$ N mm$^{-2}$. The two extreme limits of $E$ correspond to the red lines in this figure.

For the interpretation of the experimental results regarding the plate–plate pairs, the experimental curves were first smoothed to remove experimental noise and were elaborated according to a mechanical model consisting of two rectangular beams joined by a flexible rotational spring at the midspan (figure 8*b*). Accordingly, the relationship between the applied force and the measured deflection can be computed as $v = F\ell^3/48EI + F\ell^2/16bK_s$, where $K_s = M/b\phi = F\ell/4b\phi$ is the secant stiffness of the flexible spring and $\phi$ is the spring rotation angle, resulting in $K_s = F\ell^2/16b(v - F\ell^3/48EI)$ and $\phi = F\ell/4bK_s$.

This experiment provided a variable value of $K_s$ that depends on the relative rotation $\phi$ between the two plates constituting the specimen. The $K_s$ versus $\phi$ curves are reported in black in figure 8*c*. The dotted red curves in this figure represent two parameterizations of the experimental curves by the following bi-exponential law:

$$K_s(\phi) = \begin{cases} K_{so} + (K_{sp} - K_{so})\exp[a_1(\phi - \phi_p)^{\alpha_1}] & \text{if } \phi \leq \phi_p \\ K_{sp}\exp[a_2(\phi - \phi_p)^{\alpha_2}] & \text{if } \phi > \phi_p \end{cases} \tag{2.3}$$

where $K_{so}$ and $K_{sp}$ are the initial and peak values of secant stiffness, $\phi_p$ is the rotation at peak, $a_1, a_2, \alpha_1$ and $\alpha_2$ are numerical parameters that determine the shape of the hardening and softening branches. The values adopted for the constitutive parameters in the bi-exponential law are $K_{so} = 0.14 \div 1.5$ Nmm mm$^{-1}$, $K_{sp} = 18.1 \div 35.4$ Nmm mm$^{-1}$, $\phi_p = 0.068 \div 0.11$ rad, $a_1 = -2500 \div -500$, $a_2 = -32 \div -20$, $\alpha_1 = 2$ and $\alpha_2 = 1.2 \div 1.5$.

The secant stiffness of the sutures can be used to obtain the bending moment–rotation relationship as $M(\phi) = K_s(\phi)\phi$. Using the bi-exponential law, the tangent stiffness of the suture can be computed as $K_t = \partial M(\phi)/\partial \phi = \partial [\bar{K}_s(\phi)\phi]/\partial \phi$, which provides

$$K_t = \begin{cases} K_{so} + (K_{sp} - K_{so})[1 + a_1\alpha_1(\phi - \phi_p)^{\alpha_1 - 1}]\exp[a_1(\phi - \phi_p)^{\alpha_1}] & \text{if } \phi \leq \phi_p \\ K_{sp}[1 + a_2\alpha_2(\phi - \phi_p)^{\alpha_2 - 1}]\exp[a_2(\phi - \phi_p)^{\alpha_2}] & \text{if } \phi > \phi_p \end{cases}. \tag{2.4}$$

A comparison between this parametric expression of $K_t$ and that obtained from the experimental measure is diagrammed in figure 8d. Here, black curves represent the experimental results, and the dotted red curves are two of the corresponding parameterizations.

These experiments have shown that, differently from plates, which have a linear elastic behaviour until brittle failure, sutures exhibit a toughening behaviour typically linked to tessellated structures [4] in which hierarchical microstructures induce energy dissipating mechanisms at different scales [57,58]. They allow for a significant relative rotation between plates by transmitting small bending moments but avoiding large rotations by a stiffening effect. The transferred bending moment rapidly grows when the relative rotations between plates increase above 0.04 radians. Damage of the suture reduces again the transferred bending moment for large relative rotation between plates.

Notably, our experiments show a weak effect of K$^+$ on the collagenous sutural ligaments. The increase of the rotational stiffness produced by K$^+$ is observed only on average and is relatively small if compared to the variation of stiffness among samples. Nonetheless, the constitutive parameters of sutures treated with K$^+$ exhibited less disperse values.

## 2.3. Photogrammetry

To obtain the overall geometry of a real echinoid test and reconstruct a geometric three-dimensional model to be analysed by the finite-element (FE) method, an intact test was photographed by using a Panasonic Lumix FZ1000 digital camera from a lateral, superior and inferior perspective, while rotating the sample on a support plane. To perform this photographic survey, the camera was mounted on a tripod, with the optical axis orthogonal to the support plane of the echinoid test, at a fixed distance of approximately 100 mm from the sample. All digital photographs, for a total of 301 shots, included a 1 mm scale for calibration. They were subsequently uploaded to a personal computer and measured using IMAGEJ®. Finally, photographs were used to reconstruct a three-dimensional geometric model of the test using a photogrammetric reconstruction by means of the Agisoft PHOTOSCAN® software (figure 9a). To calibrate the three-dimensional model, the test of *P. lividus* was manually measured with a calliper obtaining the diameter of the test at ambitus and height. Successively, the echinoid test was cut in half along a meridian plane to measure the shell thickness at different positions. The average shell thickness was estimated to be about 1 mm. The geometry of single plates and sutures was barely visible from the photogrammetric reconstruction since these superficial features were smaller than the survey accuracy. For this reason, the size and location of sutures have been surveyed manually and, consequently, their geometry was simplified.

## 2.4. Finite-element analysis

The geometry obtained from the photogrammetric acquisition was regularized by employing a parameterized description of the shell mid-surface. The parameterized model was obtained by using CAD tools in Bentley MICROSTATION v8i. Subsequently, this surface was partitioned into plates with an irregular hexagonal shape, similar to that of the echinoid test plates (figure 8b). A thin strip of width $W_{\text{sut}} = 0.2$ mm was selected at the boundary of each plate and used to model sutures. To this end, mechanical properties of this selection was modified to reproduce the flexural behaviour of the sutures. Surfaces representing plates and sutures were meshed by employing quadrilateral shell finite elements to which a 1 mm thickness was assigned (figure 9c). The employment of shell elements for modelling both plates and sutures resulted in two perfectly equivalent models that differed only for the mechanical properties assigned to the elements associated with the sutures. In particular, the first

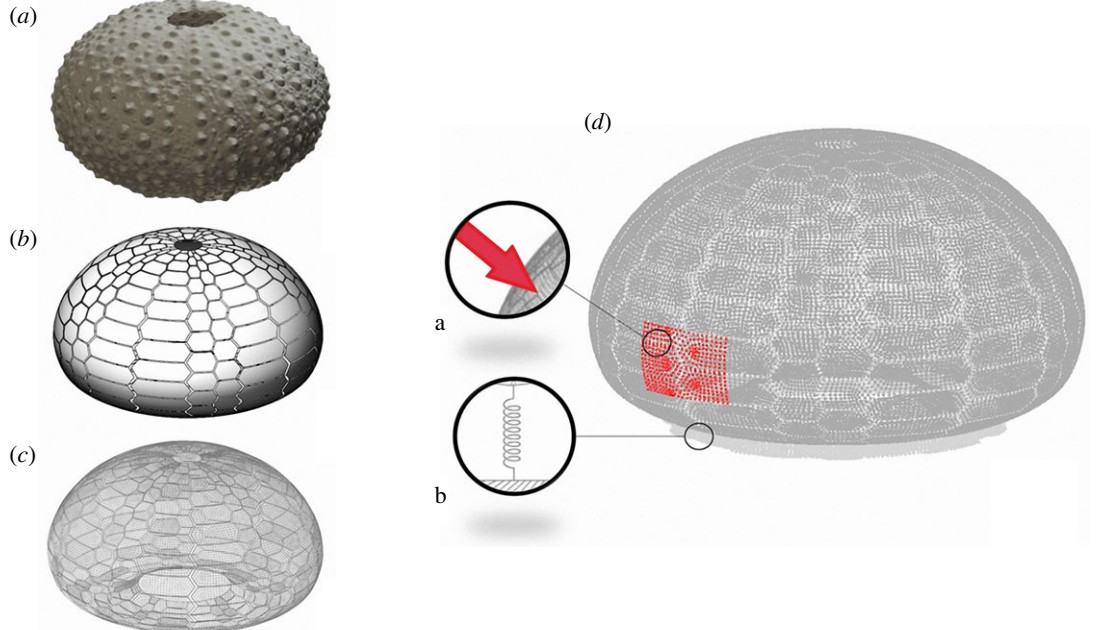

**Figure 9.** Three-dimensional reconstruction of the *P. lividus's* test. (*a*) Three-dimensional model obtained by a photogrammetric reconstruction. (*b*) Parameterized geometry model with visible plates and sutures. (*c*) Three-dimensional mesh used for finite-element analyses. (*d*) Application of pressures on a rectangular region of the test, a, and spring hinges at the base, b.

model represented a monolithic shell in which both plates and sutures had the same Young modulus, whose value was determined by the three-point bending test on single plates (see §2.2). The second model corresponded to a tessellated shell, in which sutural elements had a stiffness that reproduced the plate–plate system behaviour described in §2.2.

The relative angle between the extremities of adjacent elements that form a suture subjected to uniaxial bending moment $M$ amounts to $\phi = MW_{\mathrm{sut}}/D$, where $D = Eh^3/12(1-\nu^2)$ is the flexural rigidity of a shell element of unitary length and $W_{\mathrm{sut}}$ is the suture width. Accordingly, combining the last two formulae with the definition of rotational stiffness of the spring (2.4), we obtained:

$$K_t = \frac{M}{\phi} = \frac{D}{W_{\mathrm{sut}}} \Leftrightarrow E_{\mathrm{sut}} = \frac{12(1-\nu^2)}{h^3} W_{\mathrm{sut}} K_t. \qquad (2.5)$$

Here $E_{\mathrm{sut}}$ is an equivalent longitudinal stiffness to be applied to the flexural behaviour of elements used to model sutures. Their membrane stiffness was kept unaltered and equal to that of the plate material.

Both the monolithic and the tessellated shells were constrained at nodes in the inferior part of the echinoid test, modelling the seabed upon which the echinoid rests. To avoid concentrated reactions at supports, they were modelled as deformable springs.

Such models were analysed to compute the effects of 18 different load conditions. In particular:

  (i) uniform normal pressure of −8.2 Pa was applied to the entire shell. It accounts for the different internal celomic/external environmental pressure [18];
 (ii) lateral force of 10 N was uniformly applied to the entire echinoid test. It corresponds to half the vertical load used in [17]; and
(iii) normal pressures of resultant 10 N were applied over 16 different regions of the echinoid test, having different areas and positions with respect to the ambulacral and interambulacral plates.

Results of these analyses are documented within the supplementary material, while only the most interesting ones are here reported. They refer to the load cases in which only a portion of the echinoid test shell was loaded. In these cases, the beneficial effect of the deformable sutures in reducing the bending moments was significant.

To visualize the effect generated by the presence of flexible sutures on the reduction of bending moments within the analysed models, the generalized stress tensor $M$ associated with the bending

**Figure 10.** FEA of the *P. lividus*'s test. (*a*) FEA and response of the monolithic model with homogeneous flexural resistance with a magnified detail in (*d*). (*b*) FEA and response of the tessellated model are characterized by reduced flexural stiffness of the elements corresponding to collagenous sutures with a magnified detail in (*e*). Values represented by the chromatic scale are in N mm mm$^{-1}$ and increase from blue to red. (*c*) Diagram showing the relationship between the value of the maximum principal bending moment $M_{max}$ computed at the quadrature points of the FE model and the number of quadrature points where $M_{max}$ is attained.

moment, e.g. equation (2.1), was computed at the quadrature points of each element and extrapolated to nodes. The corresponding maximum absolute value of the principal bending moments was used to estimate the maximum bending moments that the specific load condition produced on all points of the shell mid-surface. This is defined as

$$M_{max} = \max\left(\left|\frac{M_x + M_y}{2} \mp \sqrt{\left(\frac{M_x - M_y}{2}\right)^2 + M_{xy}^2}\right|\right). \tag{2.6}$$

The contour plot of the computed values of $M_{max}$ is shown in figure 10 for both the monolithic and tessellated shell, where the same chromatic scale was used to ease comparison. In particular, such results refer to the case of a normal pressure having resultant of 10 N, applied to a rectangular portion of size 10.1 mm × 6.7 mm on the shell. Other results can be found within the electronic supplementary material and show similar behaviour. From these solutions, the significant effect that deformable sutures have on the reduction of bending moments induced by the considered loading conditions is clearly visible. The overall maximum absolute value of the principal bending moments is reduced from 0.26 Nmm mm$^{-1}$, for the monolithic shell, to 0.16 Nmm mm$^{-1}$, for the tessellated model. These peak values were attained near the centre of the loaded plates.

Furthermore, to show how flexible sutures are responsible for a reduction of the bending moments within the entire shell and not just their peak values, we report in figure 10*c* a diagram showing the

relationship between the value of the maximum principal bending moment $M_{\max}$ computed at the quadrature points of the FE model and the number of quadrature points where $M_{\max}$ is attained.

In addition to the analyses and results already here documented, we report within the electronic supplementary material: a series of additional results aiming at verifying that the capability of the flexible sutures to reduce bending moments is not specific to this particular model, but it represents a more general property of the tessellated shells. By computing the displacements associated with every considered loading condition, we verified that sutures, though responsible for an increase of the echinoid test deformability, do not produce any rigid-body mechanisms between plates. By varying the parameterization of the surveyed geometry, we verified that significant variations of the global shape of the echinoid test do not affect the capability of the flexible sutures to mitigate bending moments. Additionally, the tessellated model was further modified to obtain the following variations of the original model:

(i) all sutures were stiffened by a factor of 5. This model was used to verify that bending moments are mitigated even when sutures are stiffened by the MTC effect provided that their flexibility is still in the order of magnitude lower than that of plates;

(ii) meridian sutures were stiffened by a factor of 5 while the stiffness of the others was kept unchanged. This model was used to verify that bending moments are reduced even when the flexibility of the sutures is not uniform on the entire echinoid test; and

(iii) elastic modulus of the ambulacral plates was reduced by a factor of 5 while keeping unchanged the other ones. This model was used to verify that the variability of the plate stiffness, e.g. caused by a different thickness or the presence of pores, has negligible effects on the capability of the flexible sutures in mitigating bending moments.

These results verified the hypothesis that the tessellated configuration of the echinoid test, composed of multiple plates joined by flexible sutures, reduces bending moments and contributes to equilibrate external loads mainly by membrane forces (see §2). The three-point bending tests also revealed that sutures allow for a significant relative rotation between plates by transmitting small bending moments but avoiding large rotations owing to a stiffening effect. Besides preventing crack propagation [4], shell tessellation can be identified as an additional functional strategy that increases the structural efficiency of the endoskeleton by fostering homogeneous employment of the material along the shell thickness. In conclusion, the structural configuration of the echinoid test seems to be an optimal structural system that gathers the need to effectively withstand environmental loads as well as biological needs such as growth [23].

# 3. Functional applications of echinoid inspired principles in building construction

The mechanical study of the echinoid test revealed a high potential in transferring functional bioinspired solutions to new diverse applications. Specifically, studies from literature, briefly described in §2, together with the above presented additional results led to the identification of interesting structural working principles that can be efficiently abstracted and applied in new bioinspired building constructions.

## 3.1. Abstraction of bioinspired design principles from the echinoid test

The analyses on the *P. lividus* test provided additional knowledge about its mechanical behaviour and the biological principles behind it. As inferable from equations (2.2*a*)–(2.2*f*), low values of bending moments are associated with more efficient exploitation of material. Analyses showed that the strategic partitioning of the test into plates and flexible sutures produced a reduction of bending moments, without compromising the global stability. Accordingly, different functional principles can be abstracted and applied to design new shell structures constituted by different modules (i.e. tiled shells) able to efficiently withstand a large variety of loading conditions, namely (see table 1): (i) modular shell systems facilitate flexibility, adaptability, standardization and prefabrication [59]; (ii) hexagonal modules represent an optimized tessellation of the surface and form a continuous modular grid [60,61]; (iii) flexible joints facilitate relative rotations between plates at small values of bending moments (see §2.2); (iv) knob-like protrusions can transmit in-plane and out-of-plane shear forces between modules (see §2.1); (v) application of the trivalent vertex principle avoids global mechanisms [54]; and (vi) curved modules avoid relative rotation between plates (see §2.1).

**Table 1.** Summary of the mechanical principles that have been abstracted and transferred from the echinoid test.

| biological details | abstracted mechanical principles | emulation in shell structures |
|---|---|---|
| growth strategy-tessellated structure | fabrication process based on the addition of new elements provides flexibility, adaptability, standardization and prefabrication | modular shell system |
| pseudo-hexagonal plates | optimized surface tessellation forms a continuous modular grid by means of a limited set of simple shapes | polygonal modules |
| flexible sutures | allow relative rotations between plates associated with small values of bending moments | flexible joints |
| finger joints | allow for the transmission of in-plane and out-of-plane shear forces between plates | interlocking joints |
| trivalent vertex arrangement | avoids global mechanisms by virtue of the plate-lattice dualism | modules arranged following trivalent vertex |
| curved edges | avoid relative rotation between plates | curved modules |

## 3.2. Design of new bioinspired tiled shells: two case studies

The abstracted principles described in §3.1 were applied to the conceptual design of new shells for building constructions characterized by a discontinuous structure composed of rigid polygonal modules having curved edges and flexible joints arranged to follow the trivalent vertex principle. This biomimetic design was applied and analysed in both form-found and free-form shell structures. Both structures were loaded by static gravity loads, while seismic actions were computed by a modal response spectrum analysis according to the Eurocode 8 provisions [62,63], where both horizontal and vertical components of the elastic response spectrum were applied. To emphasize the effect produced by the presence of flexible joints between plates, both a monolithic and a tessellated model of the structures were analysed. The relevant results were compared in terms of the maximum principal bending moment computed at each quadrature point of the FE models. These comparisons were used to verify if the flexible joints were capable to reduce bending moments generated by gravity and seismic actions on medium-large-scale shell structures having a completely different geometry with respect to the echinoid test.

### 3.2.1. Case study 1: form-found shell roof

The first example regards a 10 m high concrete shell roof that covers a square area of 30 m × 30 m having a thickness of 0.3 m. The geometry of the shell mid-surface was determined by employing a finite-difference implementation of the membrane theory of shells where working stresses are given as a design parameter, whereas the shell node heights are taken as unknown. This form-finding process generated the geometry of a funicular shell, optimized to withstand its own weight [64]. The entire shell is partitioned into an arbitrary tessellation of irregular hexagonal plates whose maximum side lengths are 4.5 m. The maximum principal value of the bending moments is contour plotted in figure 11 for both monolithic and tessellated models. Specifically, these results are relevant to both horizontal and vertical seismic actions. The presence of flexible joints has an effect that depends on the direction of the seismic action with respect to the shell tessellation. The vertical component of the seismic action produces similar peak values of the maximum principal bending moment on both monolithic and tessellated shells. Thus, the presence of flexible joints has a negligible effect if the comparison is simply limited to these maximum values. However, the contour plots of figure 11 show a migration of the peak values that tend to concentrate to the centre of just a few plates. The reason for such a behaviour is certainly owing to the specific geometry of the structure, which is intentionally designed to behave efficiently when subjected to vertical loadings. On the other hand, the maximum principal values of the bending moments generated by the horizontal component of the seismic action are reduced from 4.5 kNm m$^{-1}$, attained at the base of the structure, to 3.0 kNm m$^{-1}$, attained at the centre of just four plates (only two of these plates are visible in figure 11). For both

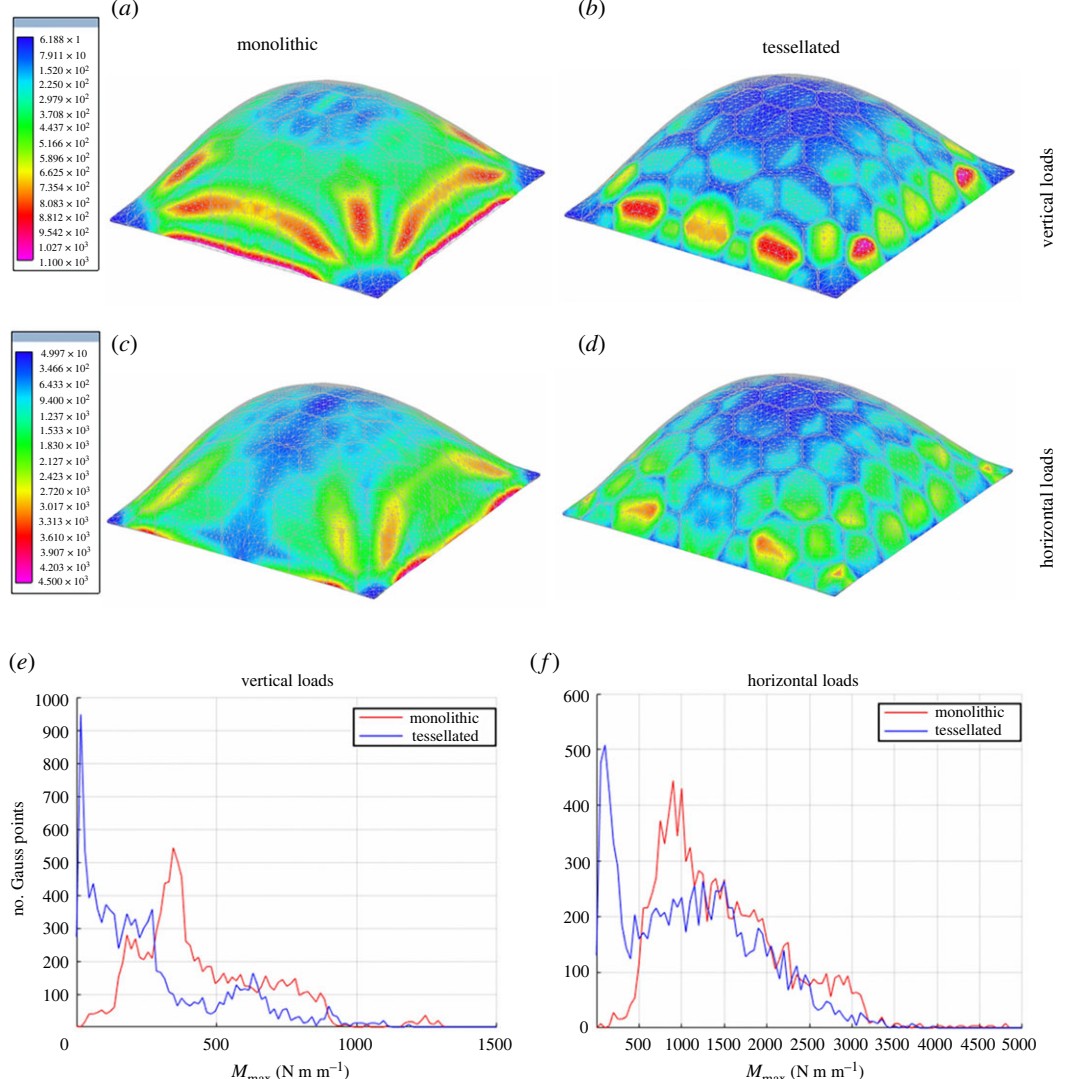

**Figure 11.** Maximum principal bending moments in funicular concrete shells: monolithic (*a,c*) versus tessellated (*b,d*); effect of vertical (*e*) and horizontal (*f*) loads. Values in the colour bars are expressed in N m m$^{-1}$. Diagrams showing the relationship between the value of the maximum principal bending moment $M_{max}$ computed at the quadrature points of the FE model and the number of quadrature points where $M_{max}$ is attained.

loading conditions, the reduction of the bending moments associated with the presence of flexible joints is clearly visible in the diagrams of figure 11, where we report the value of the maximum principal bending moment computed at the quadrature points of the FE model versus the number of quadrature points experiencing this value of the maximum principal bending moment.

### 3.2.2. Case study 2: arch shell

The second example concerns a concrete arch shell. Its mid-surface is obtained by slicing a spherical dome of radius 5 m using parallel planes placed at the distance of ±1.3 m from the centre. Differently from the previous case, the geometry of this structure is determined independently from the applied loads. The structure has a thickness equal to 0.30 m and is hinged at the base. It is partitioned into irregular pentagonal plates 1.6 m high. Since all plates have the exact same geometry, this case study can be conceived as representative of shell structures composed of prefabricated modules. The maximum principal values of the bending moments, computed for horizontal and vertical seismic loadings and for both monolithic and tiled shells, are contour plotted in figure 12. Although peak values are only slightly reduced, these tend to be confined to a small region near the vertex of a few pentagonal plates. The effect of the flexible joints in reducing the bending moments, on a great

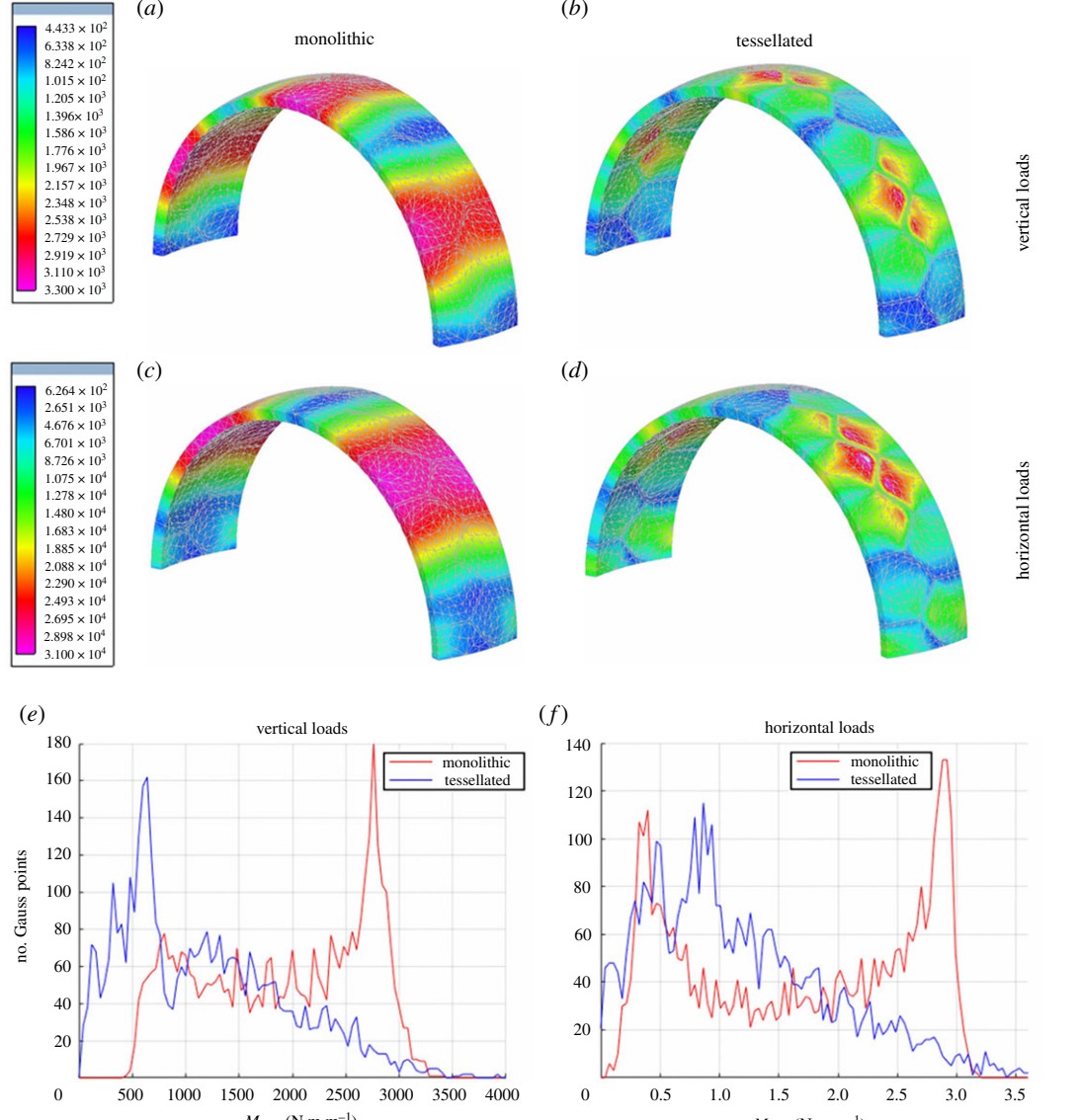

**Figure 12.** Maximum principal bending moments in concrete arch shells: monolithic ($a,c$) versus tessellated ($b,d$); effect of vertical ($e$) and horizontal ($f$) loads. Values in the colour bars are expressed in N m m$^{-1}$. Diagrams showing the relationship between the value of the maximum principal bending moment $M_{max}$ computed at the quadrature points of the FE model and the number of quadrature points where $M_{max}$ is attained.

portion of the structure, is clearly demonstrated both by colour shades in the contour plots and the diagrams of figure 12.

## 4. Conclusion

This study demonstrated how shell tessellation of the echinoid test, associated with the specific and strategic employment of suitable joints and collagenous ligaments, is a successful adaptive strategy in increasing the structural efficiency of the endoskeleton. Based on experimental observations and numerical comparisons relevant to reconstruct the mechanical behaviour of the *P. lividus* test, this research proved that shell tessellation is responsible for the reduction of bending moments in the echinoid test shell. This result was achieved by studying several aspects regarding its structural behaviour.

In particular, the research integrated and extended results consolidated in the fields of biomechanics and structural engineering, such as the increase of echinoid test strength observed by Ellers *et al.* [10], correctly attributed to the presence of collagen ligaments, and the inhibition of rigid mechanisms

described by Wester [30,31,54] for natural and engineered structures composed of plates and cylindrical hinges. In addition, a visual survey of the echinoid test showed that global and local rigid mechanisms are not only inhibited by the presence of the trivalent (Y shaped) vertices in the geometrical organization of sutures, which was deduced from Wester's kinematic analysis, but also by the in-plane and out-of-plane curvature of sutures. Hence, while relative motion between plates is avoided by the mentioned geometrical properties, micro-structured flexible sutures, composed of collagenous ligaments and knob-like protrusions, avoid separation and sliding between plates and exhibit a toughening mechanism under localized loads.

The bending flexibility of sutures was investigated by three-point bending tests, which also served to characterize the mechanical behaviour of plates. These experiments were conducted on single plates and plate–plate pairs joined by sutures. Plates have a linear elastic behaviour until brittle failure while sutures exhibit a nonlinear behaviour and are capable of allowing small relative rotations while larger rotations are inhibited by a stiffening effect. This complex behaviour was analytically described by a bi-exponential function that combines the relative rotation between plates with the rotational stiffness of sutures. Additionally, the $K^+$ effect on collagen sutures was investigated. Results on this specific point are only preliminary and show a weak effect of $K^+$ on the response of plate–plate pairs. Further experimentation is certainly needed to confirm the presence of MTC in test sutures and to characterize its behaviour.

At the global scale, the effects of the peculiar tessellated structural organization of the echinoid test were studied with the aid of a series of related FE models, analysed under the action of different loading conditions. These models take advantage of a geometrical parametrization that was used to vary the shell shape and the mechanical properties of sutures and plates, showing that the observed results are not peculiar to just one specific characteristic of the analysed model but can be generalized to a diversity of tessellated architectures.

Specific features responsible for the reduction of bending moments in tessellated shells are: (i) shell subdivision into plates and flexible sutures; (ii) curvature of the shell mid-surface; (iii) small thickness (compared to curvature); (iv) tangential and normal curvature of sutures; and (v) trivalent vertex organization of sutures. These characteristics were transferred to shells of completely different scales, shapes and materials and subjected to completely different loading conditions, such as shell roofs subject to both horizontal and vertical seismic loads. In this regard, two case studies were considered, respectively, representative of form-found and free-form shells. Numerical comparisons showed that the presence of flexible joints was responsible for a reduction of the bending moments on the entire structure. The most interesting effect regards the distribution of bending moments within the structure, which was modified by the presence of flexible joints: higher values migrated from vast diffused portions of the structure to just a few critical points. The reduction factor depended both on the structural geometry and on the direction of the applied loads. This effect is particularly visible in structures whose geometry is determined independently from the applied loads or for loading directions not considered in the form-finding analysis.

The achieved results are encouraging and provide a starting point for future research in which several aspects of both the mechanics of the echinoid test and the proposed design strategy could be further investigated including: (i) echinoid suture micromorphology variability; (ii) plates and plate–plate interaction micromechanics; (iii) unequivocal demonstration of MCT presence at sutures level and related mechanical adaptability; (iv) detailed design of flexible joints in engineered tessellated shells; (v) optimization of size and shape of shell modules; and (vi) ductility and collapse mechanisms of tessellated shell structures.

Ethics. Animal sampling was performed according to the authorization of Marina Mercantile (DPR 1639/68, 09/19/1980, confirmed by D. Lgs. 9/01/2012 n.4) and in full compliance with the European Union guidelines (directive 2010/63 and following D. Lgs. 4/03/2014 n.26). The number of animals used for experimental purposes was reduced to minimum, and only samples required to obtain reliable data were used.

Data accessibility. The datasets supporting this article have been uploaded as part of the electronic supplementary material [65].

Authors' contributions. F.M.: conceptualization, data curation, formal analysis, investigation, methodology, project administration, resources, software, supervision, validation, visualization, writing—original draft and writing—review and editing; V.P.: conceptualization, data curation, formal analysis, investigation, methodology, project administration, resources, supervision, validation, visualization, writing—original draft and writing—review and editing; A.C.: conceptualization, formal analysis, investigation, methodology, resources, supervision, validation and writing—review and editing; M.D.C.C.: conceptualization, investigation, methodology, supervision and writing—review and editing; C.L.: conceptualization, investigation, methodology, supervision and writing—review and

editing; L.R.: conceptualization, funding acquisition, investigation, methodology, project administration, resources, software, supervision and writing—review and editing.

All authors gave final approval for publication and agreed to be held accountable for the work performed therein.

Conflict of interest declaration. We declare we have no competing interests.

Funding. This work was supported by the Department of Structures for Engineering and Architecture, University of Naples Federico II, Napoli, Italy.

Acknowledgements. The Authors thank Lucas Fabian Olivero (University of Campania 'Luigi Vanvitelli'; Aversa, Italy) for the photogrammetric acquisition and reconstruction of the *P. lividus* test. They also express thanks to Sergio Bravi, University of Naples Federico II for the sample preparation; Dr Luigia Santella and Davide Caramiello the Zoological Station Anton Dohrn (Naples, Italy) for kind support, sample collection and access to the scanning electron microscope.

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
