## [Peer Review File · Royal Society Open Science]

Review History

RSOS-202003.R0 (Original submission)

Review form: Reviewer 1

Is the manuscript scientifically sound in its present form?

Yes

Are the interpretations and conclusions justified by the results?

Yes

Is the language acceptable?

Yes

Do you have any ethical concerns with this paper?

No

Have you any concerns about statistical analyses in this paper?

No

Recommendation?

Accept with minor revision (please list in comments)

Comments to the Author(s)

Tissue connections:

That tissue connections between the plates are important for the structural integrity of echinoids (they hold the skeleton together) and that they play a role for dissipating energy is somewhat of a moot point and has been emphatically stated in the older literature. For example, the structural strengthening of the regular sea urchin by sutural tissues has been explored in detailed by Ellers et al 1998, who also included general crushing complete tests as well as three-point bending experiments of connected plates of both fresh and denatured specimens. It is not clear how the present experiments differ substantially from that of Ellers et al 1998. This should be stated in more detail with respect to the experimental set up as well as results. The same holds true for the FEM analysis of Philippi & Nachtigall 1996. Again, what are the new results and how do these analyses differ?

Role of MCT:

Mutable Collagenous Tissue is a well-known feature in all echinoderm skeletons, though its specific distribution is not well known in echinoids, except for studies with respect to the jaw apparatus as well as spine attachments. First of all, not all connective tissues in the echinoderm skeleton represent MCT and it is not clear if MCT, in fact, is present between the plates of *P. lividus*. Secondly, it is not clear if this tissue in fact plays a role in stiffening the test during periods of stress. It should be more clearly stated to what extent MCT plays an integral role for the stability of the echinoids test and if the present investigations allow this to be inferred.

Plate connections:

The morphology of the sutures between different plates depends greatly on the position of the plates within the test and the specifics of which sutures are, in fact, being compared. This should be divulged in more detail. The interplate skeletal protrusions present in *P. lividus* these are certainly not comparable to the interdigitating stereomic protrusions known to be present in irregular clypeasteroid echinoids. In fact, plate connections between plates in *P. lividus* are more peg-like rounded protrusions, not the intense finger joints as implied in Figure 2. (see recent publications of Mancosu & Nebelsick 2020 *Palaeontologica Electronica* PE 23(2):a42 for SEM pictures of both peg-like protrusions as well as tissue connections between plates).

Review form: Reviewer 2

Is the manuscript scientifically sound in its present form?

No

Are the interpretations and conclusions justified by the results?

No

Is the language acceptable?

No

Do you have any ethical concerns with this paper?

No

Have you any concerns about statistical analyses in this paper?

No

Recommendation?

Major revision is needed (please make suggestions in comments)

Comments to the Author(s)

General comments:

This work uses mechanical testing and finite element modeling to illustrate the roles of different structural features in one echinoid species' mineralized test (with regard to flexural rigidity of skeleton material, segments interlocking and collagenous sutures) to argue the mechanical importance of the test's tessellated design, for the animal's ecology, but also for bioinspired design. The topic and system are relevant and timely for current interests in structured biomaterials research and sea urchins have proved a useful model system for bioinspired architecture lately. However, there are some issues with the design of the experiments and manuscript that require restructuring and/or deeper discussion:

1. Language: A comparatively minor point – the work suffers from a variety of grammatical issues, particularly subject-verb agreements (e.g. plural noun, singular verb – e.g. [page 2, line 50], [page 3, line 29]) and misspellings (even of the study species' name), but also many sentences with convoluted phrasing that is difficult to follow (I point out instances below). This is likely the result of translation, but requires more stringent polishing.
2. Scope: There is an odd mismatch between the purported scope of the paper and the actual content. Whereas the intro and even the title suggest the work will cover a much broader range of topics fundamental to how echinoid shell structures are designed in general, the core of the work is really a focused study of some specific aspects of plate-plate actions/features in a single species. It then reads as if the authors were pushing to make the work sound 'bigger' than it actually is, which only serves to dilute some nice messages here. I would find the study much more convincing if much of the extraneous 'fat' were trimmed away to center on the core research questions relating to *P. lividus* (e.g. the very broad cook's tour of ideas in the introduction could be condensed considerably and moved to an 'implications' section in the conclusions). Along these lines, it needs to be clearer why we should be invested in looking at this species, given all the work that has apparently already been done on urchins. It wasn't until [page 5, lines 3-5] that I got a sense even of the actual scope of the work and the motivation (although I am still unclear what the mentioned 'hypothesis' is).
3. Experiments: The experiments themselves are also admittedly quite limited: as far as I can tell, the mechanical tests involve only one or two samples for each condition and the biological FEA models include one monolithic model and one segmented model? On P4 20 and 49, the authors mentioned shell structures are optimized for both bending and out-of-plane shear, however, there is no support given for this statement. To explore this more, I would recommend adding shear stiffness tests and other load cases to the FE analysis. Although the authors claim the morphology is 'deeply investigated', this is presented only in quite small images in Figure 2, with most data relating to architectures that can be viewed from the surface only. Given that the mechanical behavior of the sutures is believed to be a function of both surface and 'through-thickness' interactions of the plates (e.g. collisions of the sutural 'fingers'), the presented analyses are insufficient. Images showing the finger-collisions (e.g. from microCT scans of plates loaded in in-plane shear and bending) and/or tighter relations of the morphologies to the shown loading curves would be much more convincing. Relating to the previous point: a smaller, tighter paper centered on the more focused questions of plate-plate interactions in this species, then those finescale data reinforced by the biological and bioinspired FEA data, would be an entirely interesting and useful contribution – I would recommend the authors reframe the work in this regard.
 - Loading environment: I recognize the work that goes into establishing a functional FEA model, but the load cases
4. investigated seem to be centered largely on bending, despite the service paid to other loading regimes in Fig. 1. Also, if 'skeletal material tends to increase in density in different

regions' (pg. 5) wouldn't this also be important to the mechanical behavior? And yet, in physical tests (stiffness of single plate), structure is 'imagined as homogenous and isotropic' P7 18. Incidentally, the equations discussed at the start of section 2 ('Investigation of the echinoid test') are interesting, but not really supported or explored here in any way, neither by citations nor tests to validate them. It's therefore unclear if that section is meant to be a result or is some consolidation of known theories? Currently, it feels quite 'tacked on'.

Specific comments: listed by [page number of reviewer copy, line number]

[2 42] Here immediately, but also throughout the intro, the context isn't clearly framed. A good example of this is that the study system isn't shown until Fig. 2 and even then the anatomical arrangements aren't clear and the images are small. I'd recommend a figure more like Fig. 1 of this paper (<https://www.sciencedirect.com/science/article/abs/pii/S1047847714002792>) as a first image to build context.

[2 48] I found this sentence hard to follow at first, perhaps because I had never heard of 'membrane actions' before (a concept mentioned over and over). Please define.

[2 58-59] Throughout the work, statements like this need citations for support.

[3 4] Morphology and scale are hard to picture (see comment [2 42])

[3 10] Biomimetic >> Biomimetics

[3 22] Is deformability an important factor here?

[3 27] easier to read, if you put 'neglecting' after 'while' and cut 'are neglected'. But at the end of this paragraph it would be useful to clarify WHY neglecting the MCT behavior is useful (e.g. since tissues like that are hard to mimic in engineering)

[4 14-15] Hard to follow - you're trying to outline an isostress scenario? In general, I found the terminology and goals of this section difficult to follow (see General Comment #4 above)

[4 Fig.1] (b) Check the arrow directions (V_x , V_y) for out-of-plane shear components

[4 56-7] Words missing, unclear

[4 58-60] Again, statements like these are baseless assumptions without citations (see comment [2 58-59])

[5 1] Words missing, unclear

[5 7-8] Don't understand the meaning here.

[5 12-5] As stated above, the 'hypothesis' is quite unclear, but it seems the goal is to demonstrate how 'collagenous sutures' help to distribute loads by 'membrane actions' - how can that truly be tested though if the true mutable nature of the collagenous tissue isn't captured/modeled? Also what does it mean to reduce 'bending actions'?

[5 32-4] Hard to picture without better anatomical context

[6 -10] If much of this is already known from other studies, what is the justification for exploring it more deeply here?

[6 14-5] Why not reference figure 3 here? Also, how are the actions in that figure determined? Simply intuitively or has this been explored more methodically?

[6 36-38] 'avoid' is an odd word choice here and sounds too personified. Do you mean perhaps 'restrict' or 'limit'? Also, would the nested curved plates in Fig. 3C/D even be able to rotate at all or would they tend to jam each others' movement?

[7 3-5] Nice experimental idea, hard to visualize though, I didn't understand it until later - what I really would have loved to see was a more specific demonstration of how microarchitecture related to the loading behaviors shown

[7 11] 'immersed' >> do you mean 'immersed'? (It's spelled differently in English from Latin)

[7 22-27] Already reported in previous paragraph; combine and condense

[Fig. 4] Is the blue line reaching yield?

[Fig. 5] Are these two separate experiments?

[8 21-23] So are you saying the mechanical behavior would be roughly the same without the collagen? Doesn't that go against your hypothesis? Could you dissolve the collagen and try that?

[Fig. 6C-E] Is this both the in-plane and through-thickness morphology?

[9 2] calibre >> caliper

[9 10-11] Unclear, reword

[9 19] How? Wasn't that behavior non-linear?

[9, 22] beding >> bending

[9 32] Reword, make simpler - I assume you're saying you set both joint and plate moduli to the plate modulus value? It's not really the stiffness of a 'single plate' but of the plate material, yes?

[9 41] Words missing, unclear.

[Fig 7] - what is the unit of FE analysis scale (strain / stress)? I'm confused how the test is loaded here - I don't see how you can end up with such localized stress(?) in D if the test is monolithic, unless that rounded rectangle shape is the shape of the loading zone? Is B really derived from A? I can't see plate shapes at all in the latter. Also, make C(a) and (b) bigger, hard to see.

[10 36-7] How is this shown specifically?

[10 39-45] How are these known? Just assumptions or summaries of others' data?

[11 20-24] Interesting idea, but how are the model constraints arrived on (e.g. element size/shape)? The form-finding method described is unclear to me. Also, I've never seen funicular used this way, reword.

[11 30] I don't see these effects at the centers of plates, please indicate.

[12 8] meaning 'typical' for manmade architectures with all elements similar?

[12 11] Are peak values really the best for evaluating the performance? That would be heavily influenced by outliers. What about the top X% or some aspect of stress density (stress/volume of interest)? Here you say the reduction is minimal, but in the conclusions this reduction is played up.

[12 43-4] Not enough morphological variants were explored to really make this claim, in my opinion.

Decision letter (RSOS-202003.R0)

Dear Dr PERRICONE

The Editors assigned to your paper RSOS-202003 "ECHINOID TEST AND SHELL STRUCTURES" have made a decision based on their reading of the paper and any comments received from reviewers.

Regrettably, in view of the reports received, the manuscript has been rejected in its current form. However, a new manuscript may be submitted which takes into consideration these comments.

We invite you to respond to the comments supplied below and prepare a resubmission of your manuscript. Below the referees' and Editors' comments (where applicable) we provide additional requirements. We provide guidance below to help you prepare your revision.

Please note that resubmitting your manuscript does not guarantee eventual acceptance, and we do not generally allow multiple rounds of revision and resubmission, so we urge you to make every effort to fully address all of the comments at this stage. If deemed necessary by the Editors, your manuscript will be sent back to one or more of the original reviewers for assessment. If the original reviewers are not available, we may invite new reviewers.

Please resubmit your revised manuscript and required files (see below) no later than 29-Jul-2021. Note: the ScholarOne system will 'lock' if resubmission is attempted on or after this deadline. If you do not think you will be able to meet this deadline, please contact the editorial office immediately.

Please note article processing charges apply to papers accepted for publication in Royal Society Open Science (<https://royalsocietypublishing.org/rsos/charges>). Charges will also apply to papers transferred to the journal from other Royal Society Publishing journals, as well as papers submitted as part of our collaboration with the Royal Society of Chemistry (<https://royalsocietypublishing.org/rsos/chemistry>). Fee waivers are available but must be requested when you submit your manuscript (<https://royalsocietypublishing.org/rsos/waivers>).

Thank you for submitting your manuscript to Royal Society Open Science and we look forward to receiving your resubmission. If you have any questions at all, please do not hesitate to get in touch.

on behalf of Professor Brooke Flammang (Associate Editor) and R. Kerry Rowe (Subject Editor)
openscience@royalsociety.org

Associate Editor Comments to Author (Professor Brooke Flammang):

Comments to the Author:

Both reviewers have identified a number of issues both in the execution of the experiments and the presentation of the results. Please see their their reviews and consider all points carefully. In particular, it is not clear how this work differs from previously published works. The material properties of the soft tissue have not been described in their role with clarity and the FEA model was not used to its greatest capacity in exploring the system.

Reviewer comments to Author:

Reviewer: 1

Comments to the Author(s)

Tissue connections:

That tissue connections between the plates are important for the structural integrity of echinoids (they hold the skeleton together) and that they play a role for dissipating energy is somewhat of a moot point and has been emphatically stated in the older literature. For example, the structural strengthening of the regular sea urchin by sutural tissues has been explored in detailed by Ellers et al 1998, who also included general crushing complete tests as well as three-point bending experiments of connected plates of both fresh and denatured specimens. It is not clear how the present experiments differ substantially from that of Ellers et al 1998. This should be stated in more detail with respect to the experimental set up as well as results. The same holds true for the FEM analysis of Philippi & Nachtigall 1996. Again, what are the new results and how do these analyses differ?

Role of MCT:

Mutable Collagenous Tissue is a well-known feature in all echinoderm skeletons, though its specific distribution is not well known in echinoids, except for studies with respect to the jaw apparatus as well as spine attachments. First of all, not all connective tissues in the echinoderm skeleton represent MCT and it is not clear if MCT, in fact, is present between the plates of *P. lividus*. Secondly, it is not clear if this tissue in fact plays a role in stiffening the test during periods of stress. It should be more clearly stated to what extent MCT plays an integral role for the stability of the echinoids test and if the present investigations allow this to be inferred.

Plate connections:

The morphology of the sutures between different plates depends greatly on the position of the plates within the test and the specifics of which sutures are, in fact, being compared. This should be divulged in more detail. The interplate skeletal protrusions present in *P. lividus* these are certainly not comparable to the interdigitating stereomic protrusions known to be present in irregular clypeasteroid echinoids. In fact, plate connections between plates in *P. lividus* are more peg-like rounded protrusions, not the intense finger joints as implied in Figure 2. (see recent publications of Mancosu & Nebelsick 2020 *Palaeontologica Electronica* PE 23(2):a42 for SEM pictures of both peg-like protrusions as well as tissue connections between plates).

Reviewer: 2

Comments to the Author(s)

General comments:

This work uses mechanical testing and finite element modeling to illustrate the roles of different structural features in one echinoid species' mineralized test (with regard to flexural rigidity of skeleton material, segments interlocking and collagenous sutures) to argue the mechanical importance of the test's tessellated design, for the animal's ecology, but also for bioinspired design. The topic and system are relevant and timely for current interests in structured biomaterials research and sea urchins have proved a useful model system for bioinspired architecture lately. However, there are some issues with the design of the experiments and manuscript that require restructuring and/or deeper discussion:

1. Language: A comparatively minor point – the work suffers from a variety of grammatical issues, particularly subject-verb agreements (e.g. plural noun, singular verb – e.g. [page 2, line 50], [page 3, line 29]) and misspellings (even of the study species' name), but also many sentences with convoluted phrasing that is difficult to follow (I point out instances below). This is likely the result of translation, but requires more stringent polishing.

2. Scope: There is an odd mismatch between the purported scope of the paper and the actual content. Whereas the intro and even the title suggest the work will cover a much broader range of topics fundamental to how echinoid shell structures are designed in general, the core of the work is really a focused study of some specific aspects of plate-plate actions/features in a single species. It then reads as if the authors were pushing to make the work sound 'bigger' than it actually is, which only serves to dilute some nice messages here. I would find the study much more convincing if much of the extraneous 'fat' were trimmed away to center on the core research questions relating to *P. lividus* (e.g. the very broad cook's tour of ideas in the introduction could be condensed considerably and moved to an 'implications' section in the conclusions). Along these lines, it needs to be clearer why we should be invested in looking at this species, given all the work that has apparently already been done on urchins. It wasn't until [page 5, lines 3-5] that I got a sense even of the actual scope of the work and the motivation (although I am still unclear what the mentioned 'hypothesis' is).

3. Experiments: The experiments themselves are also admittedly quite limited: as far as I can tell, the mechanical tests involve only one or two samples for each condition and the biological FEA models include one monolithic model and one segmented model? On P4 20 and 49, the authors mentioned shell structures are optimized for both bending and out-of-plane shear, however, there is no support given for this statement. To explore this more, I would recommend adding shear stiffness tests and other load cases to the FE analysis. Although the authors claim the morphology is 'deeply investigated', this is presented only in quite small images in Figure 2, with most data relating to architectures that can be viewed from the surface only. Given that the mechanical behavior of the sutures is believed to be a function of both surface and 'through-thickness' interactions of the plates (e.g. collisions of the sutural 'fingers'), the presented analyses are insufficient. Images showing the finger-collisions (e.g. from microCT scans of plates loaded in in-plane shear and bending) and/or tighter relations of the morphologies to the shown loading curves would be much more convincing. Relating to the previous point: a smaller, tighter paper centered on the more focused questions of plate-plate interactions in this species, then those finescale data reinforced by the biological and bioinspired FEA data, would be an entirely interesting and useful contribution – I would recommend the authors reframe the work in this regard.

- Loading environment: I recognize the work that goes into establishing a functional FEA model, but the load cases

4. investigated seem to be centered largely on bending, despite the service paid to other loading regimes in Fig. 1. Also, if 'skeletal material tends to increase in density in different regions' (pg. 5) wouldn't this also be important to the mechanical behavior? And yet, in physical tests (stiffness of single plate), structure is 'imagined as homogenous and isotropic' P7 18. Incidentally, the

equations discussed at the start of section 2 ('Investigation of the echinoid test') are interesting, but not really supported or explored here in any way, neither by citations nor tests to validate them. It's therefore unclear if that section is meant to be a result or is some consolidation of known theories? Currently, it feels quite 'tacked on'.

Specific comments: listed by [page number of reviewer copy, line number]

[2 42] Here immediately, but also throughout the intro, the context isn't clearly framed. A good example of this is that the study system isn't shown until Fig. 2 and even then the anatomical arrangements aren't clear and the images are small. I'd recommend a figure more like Fig. 1 of this paper (<https://www.sciencedirect.com/science/article/abs/pii/S1047847714002792>) as a first image to build context.

[2 48] I found this sentence hard to follow at first, perhaps because I had never heard of 'membrane actions' before (a concept mentioned over and over). Please define.

[2 58-59] Throughout the work, statements like this need citations for support.

[3 4] Morphology and scale are hard to picture (see comment [2 42])

[3 10] Biomimetic >> Biomimetics

[3 22] Is deformability an important factor here?

[3 27] easier to read, if you put 'neglecting' after 'while' and cut 'are neglected'. But at the end of this paragraph it would be useful to clarify WHY neglecting the MCT behavior is useful (e.g. since tissues like that are hard to mimic in engineering)

[4 14-15] Hard to follow - you're trying to outline an isostress scenario? In general, I found the terminology and goals of this section difficult to follow (see General Comment #4 above)

[4 Fig.1] (b) Check the arrow directions (V_x , V_y) for out-of-plane shear components

[4 56-7] Words missing, unclear

[4 58-60] Again, statements like these are baseless assumptions without citations (see comment [2 58-59])

[5 1] Words missing, unclear

[5 7-8] Don't understand the meaning here.

[5 12-5] As stated above, the 'hypothesis' is quite unclear, but it seems the goal is to demonstrate how 'collagenous sutures' help to distribute loads by 'membrane actions' - how can that truly be tested though if the true mutable nature of the collagenous tissue isn't captured/modeled? Also what does it mean to reduce 'bending actions'?

[5 32-4] Hard to picture without better anatomical context

[6 -10] If much of this is already known from other studies, what is the justification for exploring it more deeply here?

[6 14-5] Why not reference figure 3 here? Also, how are the actions in that figure determined? Simply intuitively or has this been explored more methodically?

[6 36-38] 'avoid' is an odd word choice here and sounds too personified. Do you mean perhaps 'restrict' or 'limit'? Also, would the nested curved plates in Fig. 3C/D even be able to rotate at all or would they tend to jam each others' movement?

[7 3-5] Nice experimental idea, hard to visualize though, I didn't understand it until later – what I really would have loved to see was a more specific demonstration of how microarchitecture related to the loading behaviors shown

[7 11] 'immersed' >> do you mean 'immersed'? (It's spelled differently in English from Latin)

[7 22-27] Already reported in previous paragraph; combine and condense

[Fig. 4] Is the blue line reaching yield?

[Fig. 5] Are these two separate experiments?

[8 21-23] So are you saying the mechanical behavior would be roughly the same without the collagen? Doesn't that go against your hypothesis? Could you dissolve the collagen and try that?

[Fig. 6C-E] Is this both the in-plane and through-thickness morphology?

[9 2] calibre >> caliper

[9 10-11] Unclear, reword

[9 19] How? Wasn't that behavior non-linear?

[9, 22] beding >> bending

[9 32] Reword, make simpler – I assume you're saying you set both joint and plate moduli to the plate modulus value? It's not really the stiffness of a 'single plate' but of the plate material, yes?

[9 41] Words missing, unclear.

[Fig 7] – what is the unit of FE analysis scale (strain / stress)? I'm confused how the test is loaded here – I don't see how you can end up with such localized stress(?) in D if the test is monolithic, unless that rounded rectangle shape is the shape of the loading zone? Is B really derived from A? I can't see plate shapes at all in the latter. Also, make C(a) and (b) bigger, hard to see.

[10 36-7] How is this shown specifically?

[10 39-45] How are these known? Just assumptions or summaries of others' data?

[11 20-24] Interesting idea, but how are the model constraints arrived on (e.g. element size/shape)? The form-finding method described is unclear to me. Also, I've never seen funicular used this way, reword.

[11 30] I don't see these effects at the centers of plates, please indicate.

[12 8] meaning 'typical' for manmade architectures with all elements similar?

[12 11] Are peak values really the best for evaluating the performance? That would be heavily influenced by outliers. What about the top X% or some aspect of stress density (stress/volume of

interest)? Here you say the reduction is minimal, but in the conclusions this reduction is played up.

[12 43-4] Not enough morphological variants were explored to really make this claim, in my opinion.

===PREPARING YOUR MANUSCRIPT===

===PREPARING YOUR REVISION IN SCHOLARONE===

<https://royalsociety.org/journals/authors/author-guidelines/#supplementary-material> to include a suitable title and informative caption. An example of appropriate titling and captioning may be found at https://figshare.com/articles/Table_S2_from_Is_there_a_trade-off_between_peak_performance_and_performance_breadth_across_temperatures_for_aerobic_sc_ope_in_teleost_fishes_/3843624.

Author's Response to Decision Letter for (RSOS-202003.R0)

See Appendix A.

RSOS-211972.R0

Review form: Reviewer 1

Is the manuscript scientifically sound in its present form?

Yes

Are the interpretations and conclusions justified by the results?

Yes

Is the language acceptable?

Yes

Do you have any ethical concerns with this paper?

Yes

Have you any concerns about statistical analyses in this paper?

No

Recommendation?

Accept with minor revision (please list in comments)

Comments to the Author(s)

Paragraph lines 180 to 204 is too long. I suggest subdivision at line 192 preceeding "Considering...."

References 4 and 47 are repeated

Review form: Reviewer 2

Is the manuscript scientifically sound in its present form?

Yes

Are the interpretations and conclusions justified by the results?

No

Is the language acceptable?

No

Do you have any ethical concerns with this paper?

No

Have you any concerns about statistical analyses in this paper?

No

Recommendation?

Reject

Comments to the Author(s)

General comments:

As previously, I greatly appreciate the topic of this research and the authors' efforts to draw on a relevant biological example for understanding the performance of tessellations in architecture. The authors have clearly taken many reviewers' critiques to heart, particularly evident in the expanded experiments and modified figures, the latter much more intuitive (although still requiring some tweaks, in my opinion; see below). Unfortunately, however, I still often found the prose overly dense and repetitive, which makes it very challenging to read, and therefore to access the nice science here. I'm not sure if this stems from disciplinary or language differences or something else, but I think having an outside editor would be invaluable: I would recommend the authors at least ask a colleague from another discipline to read the work and highlight text that is particularly unclear or confusing (I point out some areas I found difficult below). I believe some steps can be taken to make headway in this regard:

1. Often, phrasing feels as if it has been translated from another language, e.g.
 135: "an optimized shell structure is capable to equilibrate"
 299 and 401: "...are capable to allow"
 333: "sutures permitted to obtain"
 520: "which resulted to be sensibly modified"
 ...such errors would be easily caught by a proof-reader
2. A minor point: I regularly tripped over the word 'test' – although it is the correct term for this shell, its other meaning is so common (especially in science) as to be a stumbling block. Perhaps 'echinoid test'?
3. Some information was noticeably repeated, e.g.
 - the background info from Ellers et al. and Philippi and Nachtigall (54-58, 151-155) – the discussion of this lit is very useful and appears to be in response to the other reviewer's comments, but will be more powerful if condensed
 - the extensive experimental summary, starting line 75, 'gives away' too many of the approaches and findings, immediately prior to the whole investigations section – this summary could be condensed by ~60% to provide a nice lead-in to the experimental section
 - the Conclusion, while well-written, rehashes much of the discussion of results already found in individual "results" sections
4. Regardless of experimental details, a basic comprehension of the findings hinges on an understanding of some specific anatomical features and some engineering/mechanics concepts. With regard to the former, it would be useful to label all the features from the 'Visual survey' in Figure 1, I had trouble relating the text to the image. With regard to the latter, defining terms like membrane forces, bending moment, mid-surface, tessellation and shell structures early in the text would be very helpful (e.g. the shell definitions on 99-101 are needed earlier).
5. The structure of the work is atypical – each 'investigation' section has its own methods, results and discussion (the latter rehashed in the conclusions, see #3 above). This can certainly work, but perhaps having a more standardized organization and more apt titling of sections would help the reader more quickly understand the format. For example, the 'Visual survey' section included an anatomical description (although the photogrammetric analysis came later, oddly), but then also a discussion of MCT and a structure-mechanics analysis. Perhaps it would help to organize the investigations by size scale (nicely done in Fig. 3), e.g. starting with general anatomy, then plate ultrastructures, then plate anatomical interactions, then plate and joint mechanics...
6. I would argue that, in general, the prose needs to be tightened and made more accessible throughout. Again, an external reviewer/editor would be extremely useful here; I know that after a while, I certainly can become 'too close' to my writing and not see extraneous text! I could easily imagine the manuscript being shortened by ~30% and it would be far stronger for it.

Specific comments: listed by [line number]

(PLEASE NOTE: up to line 150, I detail all comments I had, to give a sense of the scope of what I saw as regular critiques for the manuscript – many of these I imagine would be fixed through careful proof-reading by an external editor; after line 150, I highlight only comparatively larger issues)

[5] Limitless is not accurate □ On the other hand ‘several’ is also used incorrectly (it should be used to main >2, but not many) on lines 31 and 50

[18-20] Does an efficient structure necessarily effectively protect human activities?

[22-23] Isn’t structural efficiency always related to the distribution of internal forces, not just in shelled structures?

[32-36] Is this needed? Why not go right into tessellations?

[48-49] What do you mean ‘expand’ – grow?

[50-54] Repetition of the strengthening action concept, even within these few lines. But the bit from 50-52, in my opinion, is one of the big keys that distinguishes your work (it just gets lost in all the other piles of information here, unfortunately)

[74] Odd wording: “effect...on the value of bending moments” (if I get what you mean, could probably just remove “the value of”)

[76] What do you mean “avoids rigid mechanisms”?

[88] Odd wording: “after recalling the advantages of designing”

[102-103] Showing these force directions on one of your figures would be useful

[137] “to make it funicular” – you use this phrasing several times, perhaps it is a normal phrasing in architecture or engineering? I have never heard it before and it reads oddly

[143-144] Careful, just because a morphology evolved, doesn’t mean it is the most optimized for weight (it could be optimized for something else, most likely for multiple, even competing factors)

[146] Support metabolic rate statement with citation

[150-151] A key point, but the logic leap to get here is not intuitive, maybe because the terms aren’t clearly defined. Yet, in the following sentence, these physical phenomena/structural behaviors are taken as fact.

[160-167] How do you know this treatment didn’t alter structure? Which plates were examined? (Relates to the question of local variation in morphology)

[180-186] Hard to follow descriptions – clear labels in figure would help

[201-203] Reword, hard to follow (also 306, 328-331, 368-370)

[208-224] This is more like a typical ‘discussion’ – see comment #5 above

[220] This and other discussions of the trivalent pattern in the text are hard to follow

[226] For the later joint loading tests, were plate-plate samples cut to avoid the jamming behavior you’re describing for curved tiles?

[261-272] Key points about the behavior of tiles vs tile-joint composites, but this could be clearer – despite the discussion typically being wrapped in with the results in your paper format, there isn’t enough context here. Explain how the stress-strain curves illustrate what you’re describing in 261-264. How is the rotation demonstrated, is this shown in any figure? “It turns out...” is very casual wording. Why is the modulus written with a division sign? Is this modulus reasonable for the materials studied? There must be some precedent... etc.

[297-302] The attribution of mechanics to structural aspects is made without clear support (e.g. is “damage of the suture” shown?). Also, the curves/trends in Fig. 6-7 are very difficult to see.

[325-328] Incorrect numbering (should be 2.4). How was the parameterized model built?

[349-354] Nice range of experiments, but they feel a bit random – e.g. what relevance do the lateral forces and normal pressures have?

[383-384] Again, rigid-body mechanisms need to be defined

[400-407] Interesting points, but aren’t all backed up with experimental data. E.g., I have yet to see demonstration of these rotations? Maybe I’m missing them in a figure? Also, I thought the

material along the thickness wasn't homogeneous? And what evidence do you have to say the shell is an optimal compromise between mechanical support and growth?

[418] If this was in the equations in 2a-2f, this could be underlined more clearly for non-mathematicians...

[424-427] Section numbering is wrong here as well and the wording of #5 is hard to follow.

[511] Again, I'd avoid dramatic words like "vast" □ Maybe just "...generalized to a diversity of tessellated architectures"?

[Fig 3] This figure looks great, nice job – relating to the whole test would be useful, maybe with a little icon of the shell? What is 'ao'? Scale bars in the images would help

[Fig 4] Nice looking images, but it's a bit hard to relate all the SEM images to the anatomies in A and to each other (e.g. it took me a moment to understand how C and D relate)

[Fig 9] If the structure is modeled with homogeneous material properties, why are the stresses confined to one tile? Was the applied load exactly the size of the tile? What are the quadrature points?

Decision letter (RSOS-211972.R0)

Dear Dr Perricone,

On behalf of the Editors, we are pleased to inform you that your Manuscript RSOS-211972 "FLEXIBLE SUTURES REDUCE BENDING MOMENTS IN SHELLS: FROM THE ECHINOID TEST TO TESSELLATED SHELL STRUCTURES" has been accepted for publication in Royal Society Open Science subject to minor revision in accordance with the referees' reports. Please find the referees' comments along with any feedback from the Editors below my signature.

Please submit your revised manuscript and required files (see below) no later than 7 days from today's (ie 07-Mar-2022) date. Note: the ScholarOne system will 'lock' if submission of the revision is attempted 7 or more days after the deadline. If you do not think you will be able to meet this deadline please contact the editorial office immediately.

Kind regards,
Royal Society Open Science Editorial Office

on behalf of Professor Brooke Flammang (Associate Editor) and R. Kerry Rowe (Subject Editor)
 openscience@royalsociety.org

Associate Editor Comments to Author (Professor Brooke Flammang):

Both reviewers have noted that significant changes have been made to the manuscript. However, Reviewer 2 still makes a number of very useful suggestions for improving the readability of this work for the very broad readership of RSOS. Please adhere to these suggestions.

Reviewer comments to Author:

Reviewer: 1

Comments to the Author(s)

Paragraph lines 180 to 204 is too long. I suggest subdivision at line 192 preceding "Considering..."

References 4 and 47 are repeated

Reviewer: 2

Comments to the Author(s)

General comments:

As previously, I greatly appreciate the topic of this research and the authors' efforts to draw on a relevant biological example for understanding the performance of tessellations in architecture. The authors have clearly taken many reviewers' critiques to heart, particularly evident in the expanded experiments and modified figures, the latter much more intuitive (although still requiring some tweaks, in my opinion; see below). Unfortunately, however, I still often found the prose overly dense and repetitive, which makes it very challenging to read, and therefore to access the nice science here. I'm not sure if this stems from disciplinary or language differences or something else, but I think having an outside editor would be invaluable: I would recommend the authors at least ask a colleague from another discipline to read the work and highlight text that is particularly unclear or confusing (I point out some areas I found difficult below). I believe some steps can be taken to make headway in this regard:

1. Often, phrasing feels as if it has been translated from another language, e.g.

135: "an optimized shell structure is capable to equilibrate"

299 and 401: "...are capable to allow"

333: "sutures permitted to obtain"

520: "which resulted to be sensibly modified"

...such errors would be easily caught by a proof-reader

2. A minor point: I regularly tripped over the word 'test' - although it is the correct term for this shell, its other meaning is so common (especially in science) as to be a stumbling block. Perhaps 'echinoid test'?

3. Some information was noticeably repeated, e.g.

- the background info from Ellers et al. and Philippi and Nachtigall (54-58, 151-155) - the discussion of this lit is very useful and appears to be in response to the other reviewer's comments, but will be more powerful if condensed

- the extensive experimental summary, starting line 75, 'gives away' too many of the approaches and findings, immediately prior to the whole investigations section - this summary could be condensed by ~60% to provide a nice lead-in to the experimental section

- the Conclusion, while well-written, rehashes much of the discussion of results already found in individual “results” sections

4. Regardless of experimental details, a basic comprehension of the findings hinges on an understanding of some specific anatomical features and some engineering/mechanics concepts. With regard to the former, it would be useful to label all the features from the ‘Visual survey’ in Figure 1, I had trouble relating the text to the image. With regard to the latter, defining terms like membrane forces, bending moment, mid-surface, tessellation and shell structures early in the text would be very helpful (e.g. the shell definitions on 99-101 are needed earlier).

5. The structure of the work is atypical – each ‘investigation’ section has its own methods, results and discussion (the latter rehashed in the conclusions, see #3 above). This can certainly work, but perhaps having a more standardized organization and more apt titling of sections would help the reader more quickly understand the format. For example, the ‘Visual survey’ section included an anatomical description (although the photogrammetric analysis came later, oddly), but then also a discussion of MCT and a structure-mechanics analysis. Perhaps it would help to organize the investigations by size scale (nicely done in Fig. 3), e.g. starting with general anatomy, then plate ultrastructures, then plate anatomical interactions, then plate and joint mechanics...

6. I would argue that, in general, the prose needs to be tightened and made more accessible throughout. Again, an external reviewer/editor would be extremely useful here; I know that after a while, I certainly can become ‘too close’ to my writing and not see extraneous text! I could easily imagine the manuscript being shortened by ~30% and it would be far stronger for it.

Specific comments: listed by [line number]

(PLEASE NOTE: up to line 150, I detail all comments I had, to give a sense of the scope of what I saw as regular critiques for the manuscript – many of these I imagine would be fixed through careful proof-reading by an external editor; after line 150, I highlight only comparatively larger issues)

[5] Limitless is not accurate □ On the other hand ‘several’ is also used incorrectly (it should be used to main >2, but not many) on lines 31 and 50

[18-20] Does an efficient structure necessarily effectively protect human activities?

[22-23] Isn’t structural efficiency always related to the distribution of internal forces, not just in shelled structures?

[32-36] Is this needed? Why not go right into tessellations?

[48-49] What do you mean ‘expand’ – grow?

[50-54] Repetition of the strengthening action concept, even within these few lines. But the bit from 50-52, in my opinion, is one of the big keys that distinguishes your work (it just gets lost in all the other piles of information here, unfortunately)

[74] Odd wording: “effect...on the value of bending moments” (if I get what you mean, could probably just remove “the value of”)

[76] What do you mean “avoids rigid mechanisms”?

[88] Odd wording: “after recalling the advantages of designing”

[102-103] Showing these force directions on one of your figures would be useful

[137] “to make it funicular” – you use this phrasing several times, perhaps it is a normal phrasing in architecture or engineering? I have never heard it before and it reads oddly

[143-144] Careful, just because a morphology evolved, doesn’t mean it is the most optimized for weight (it could be optimized for something else, most likely for multiple, even competing factors)

[146] Support metabolic rate statement with citation

[150-151] A key point, but the logic leap to get here is not intuitive, maybe because the terms aren’t clearly defined. Yet, in the following sentence, these physical phenomena/structural behaviors are taken as fact.

- [160-167] How do you know this treatment didn't alter structure? Which plates were examined? (Relates to the question of local variation in morphology)
- [180-186] Hard to follow descriptions – clear labels in figure would help
- [201-203] Reword, hard to follow (also 306, 328-331, 368-370)
- [208-224] This is more like a typical 'discussion' – see comment #5 above
- [220] This and other discussions of the trivalent pattern in the text are hard to follow
- [226] For the later joint loading tests, were plate-plate samples cut to avoid the jamming behavior you're describing for curved tiles?
- [261-272] Key points about the behavior of tiles vs tile-joint composites, but this could be clearer – despite the discussion typically being wrapped in with the results in your paper format, there isn't enough context here. Explain how the stress-strain curves illustrate what you're describing in 261-264. How is the rotation demonstrated, is this shown in any figure? "It turns out..." is very casual wording. Why is the modulus written with a division sign? Is this modulus reasonable for the materials studied? There must be some precedent... etc.
- [297-302] The attribution of mechanics to structural aspects is made without clear support (e.g. is "damage of the suture" shown?). Also, the curves/trends in Fig. 6-7 are very difficult to see.
- [325-328] Incorrect numbering (should be 2.4). How was the parameterized model built?
- [349-354] Nice range of experiments, but they feel a bit random – e.g. what relevance do the lateral forces and normal pressures have?
- [383-384] Again, rigid-body mechanisms need to be defined
- [400-407] Interesting points, but aren't all backed up with experimental data. E.g., I have yet to see demonstration of these rotations? Maybe I'm missing them in a figure? Also, I thought the material along the thickness wasn't homogeneous? And what evidence do you have to say the shell is an optimal compromise between mechanical support and growth?
- [418] If this was in the equations in 2a-2f, this could be underlined more clearly for non-mathematicians...
- [424-427] Section numbering is wrong here as well and the wording of #5 is hard to follow.
- [511] Again, I'd avoid dramatic words like "vast" □ Maybe just "...generalized to a diversity of tessellated architectures"?
- [Fig 3] This figure looks great, nice job – relating to the whole test would be useful, maybe with a little icon of the shell? What is 'ao'? Scale bars in the images would help
- [Fig 4] Nice looking images, but it's a bit hard to relate all the SEM images to the anatomies in A and to each other (e.g. it took me a moment to understand how C and D relate)
- [Fig 9] If the structure is modeled with homogeneous material properties, why are the stresses confined to one tile? Was the applied load exactly the size of the tile? What are the quadrature points?

===PREPARING YOUR MANUSCRIPT===

- one version should clearly identify all the changes that have been made (for instance, in coloured highlight, in bold text, or tracked changes);
- a 'clean' version of the new manuscript that incorporates the changes made, but does not highlight them. This version will be used for typesetting.

===PREPARING YOUR REVISION IN SCHOLARONE===

-- If you are requesting an article processing charge waiver, you must select the relevant waiver option (if requesting a discretionary waiver, the form should have been uploaded, see 'File upload' above).

-- If you have uploaded any electronic supplementary (ESM) files, please ensure you follow the guidance at <https://royalsociety.org/journals/authors/author-guidelines/#supplementary-material> to include a suitable title and informative caption. An example of appropriate titling and captioning may be found at https://figshare.com/articles/Table_S2_from_Is_there_a_trade-off_between_peak_performance_and_performance_breadth_across_temperatures_for_aerobic_scope_in_teleost_fishes_/3843624.

Author's Response to Decision Letter for (RSOS-211972.R0)

See Appendix B.

Decision letter (RSOS-211972.R1)

Dear Dr Perricone,

I am pleased to inform you that your manuscript entitled "FLEXIBLE SUTURES REDUCE BENDING MOMENTS IN SHELLS: FROM THE ECHINOID TEST TO TESSELLATED SHELL STRUCTURES" is now accepted for publication in Royal Society Open Science.

on behalf of Professor Brooke Flammang (Associate Editor) and R. Kerry Rowe (Subject Editor)
openscience@royalsociety.org

Appendix A

Response to reviewers

We gratefully thank both reviewers for their accurate revision of the manuscript. Their concerns have been carefully considered and addressed as follows:

Reviewer 1

Tissue connections:

That tissue connections between the plates are important for the structural integrity of echinoids (they hold the skeleton together) and that they play a role for dissipating energy is somewhat of a moot point and has been emphatically stated in the older literature. For example, the structural strengthening of the regular sea urchin by sutural tissues has been explored in detailed by Ellers et al 1998, who also included general crushing complete tests as well as three-point bending experiments of connected plates of both fresh and denatured specimens. It is not clear how the present experiments differ substantially from that of Ellers et al 1998. This should be stated in more detail with respect to the experimental set up as well as results. The same holds true for the FEM analysis of Philippi & Nachtigall 1996. Again, what are the new results and how do these analyses differ?

Thank you for this observation. The study carried out by Ellers et al. (1998) on *S. purpuratus* was extremely inspiring and guided our work on *P. lividus* confirming the importance of the sutures in the mechanical design of regular echinoid test. Primarily, our study differs from Ellers et al. (1998) for the goals of our experiments. Our work was intended to investigate the bending behaviour of sutures and how this influences the distribution of bending moments within the entire shell of the echinoid test. Conversely, the experiments carried out by Ellers et al. are mainly focused on the strength of sutures. For this reason, although the experimental setup of our the three-point bending tests is similar to that made by Ellers et al., the experimental output is finalized to the determination of the stiffness-rotation relationship of sutures in order to identify the relevant mechanical parameters (i.e., see the determination of K_s and K_t in section 3.2, equations (3) and (4) and figures 6 and 7). Accordingly, our FE-Analyses are focused on the role that the high flexibility of sutures has on the distribution of bending moments within the test shell structure. This phenomenon has also been neglected by Philippi and Nachtigall (1996): their analyses were focused on studying the mechanical behaviour response of the *E. esculentus* test to external loads considering it as a monolithic model and thus neglecting its division into modules and the flexibility of sutures. Additionally, their model was obtained by replicating a slide corresponding to one tenth of the entire shell. This was possible thanks to the double pentagonal symmetry of the structure. However, modelling just one sector of the entire shell implicitly imposes that also loading conditions fulfil the same symmetry, which is not the case of our analyses. The loading condition shown in figure 8 is only symmetric about a meridian plane, and this plane does not coincide with any of the ten symmetry planes for the test. Also, as detailed in the revised version of the manuscript, a total of 18 different load cases have been considered in our study and they do not take advantage of the symmetry of the structure. Additionally, we have modified the geometry and the mechanical properties of the test shell in order to properly generalize our results.

Based on these reviewer's comments, we extended the discussion regarding previous studies in the introduction (**lines 50-58**) and in conclusions (**lines 488-511**) highlighting similarities and differences with respect to the mentioned research works.

Role of MCT:

Mutable Collagenous Tissue is a well-known feature in all echinoderm skeletons, though its specific distribution is not well known in echinoids, except for studies with respect to the jaw apparatus as well as spine attachments. First of all, not all connective tissues in the echinoderm skeleton represent MCT and it is not clear if MCT, in fact, is present between the plates of P. lividus. Secondly, it is not clear if this tissue in fact plays a role in stiffening the test during periods of stress. It should be more clearly stated to what extent MCT plays an integral role for the stability of the echinoids test and if the present investigations allow this to be inferred.

We find this observation very relevant. In order to preliminary investigate the possible effects of MCT presence in *P. lividus* test, we decided to repeat all three-points bending tests considering 5 samples immersed in filtered seawater and 5 immersed in 100 mM K⁺ (a treatment usually inducing stiffening in established MCT ligaments), for a total of 10 samples. As reported in section 3.2 (**lines 303-306**) of the revised manuscript, these new experiments show a weak stiffening effect of K⁺ on the sutural ligaments. Actually, the increase of rotational stiffness produced by K⁺ is relatively small if compared to its variation among specimens. However, we also noticed that sutures treated with K⁺ exhibited less disperse values of the constitutive parameters. Our analyses are only preliminary for a dedicated mechanical study on MTC presence in *P. lividus* sutures and further analyses and test with other neuroactive agents are therefore requested to statistically confirm the presence/absence of MCT at sutures and characterize their stiffening/softening effects. We decided to dedicate an *ad hoc* investigation to assess this aspect that is negligible in this study. Actually, to estimate the role of MTC on the capability of sutures to reduce bending moments, we performed additional FE analyses in which the stiffness of sutures has been increased as if they were subjected to the stiffening effect produced by MTC in stiffened state. In other words, we assigned a bending stiffness to elements associated to sutures that is five times larger than the one associated to the original FE model (**lines 388-390, 504-506**). The corresponding results, which are reported in the supplementary material attached to the revised version of the manuscript, show that the actual stiffness of sutures has very low effect on the distribution of bending moments within the test shell, provided that elements associated to sutures have a bending stiffness that is at least one order of magnitude lower than that of plates. Noticeably, our three point bending tests have shown that the factor five, used in our analyses, is an overestimate of the actual effect of K⁺ on sutural ligaments, hence equivalent bending stiffness of sutures is practically always about two orders of magnitude lower than plates. Hence, even if MTC was present in *P. lividus* test, its effect on the capability of sutures to mitigate bending moments appears to be negligible.

In conclusion, although the presence and effects of MTC in *P. lividus* test sutures is an interesting aspect which we are currently planning to investigate by developing new experiments, we believe this effect is inessential to the central goal of the present manuscript thereby deciding to leave any further investigation on this issue to future research.

Plate connections:

The morphology of the sutures between different plates depends greatly on the position of the plates within the test and the specifics of which sutures are, in fact, being compared. This should be divulged in more detail. The interplate skeletal protrusions present in P. lividus these are certainly not comparable to the interdigitating stereomic protrusions known to be present in irregular clypeasteroid echinoids. In fact, plate connections between plates in P. lividus are more peg-like rounded protrusions, not the intense finger joints as implied in Figure 2. (see recent publications

of Mancosu & Nebelsick 2020 Palaeontologica Electronica PE 23(2):a42 for SEM pictures of both peg-like protrusions as well as tissue connections between plates).

Thank you for pointing out this aspect. In all our experiments, we tested and modelled exclusively interradial sutures present at midline of the interambulacral zone. This is due to the objective difficulty of sampling sufficiently large specimens that include other types of sutures. However, the effect of suture morphology can be accounted in FE analyses by correcting the bending stiffness of corresponding elements. Following the approach described in our previous answer regarding the role of MTC, we increased the bending stiffness of interradial sutures leaving unaltered the stiffness of circumferential sutures. These results are mentioned in the revised version of the manuscript (**lines 391-393**) and reported in more detail within the attached supplementary material. The variability of suture stiffness has negligible effects on the capability of sutures to mitigate bending moments into the test.

Regarding the description of interplate skeletal protrusions, the interradial sutures of the interambulacral zone are characterized by several rounded protrusions, which are quite intense as visible in the new figure 3 and according to the description of Mancosu and Nebelsick (2020). In collaboration with A. Mancosu and J. Nebelsick, we are currently carrying out an in-depth morphological analysis of the different suture in *P.lividus* using SEM micrographs and Micro-CT scans. As this reviewer points out, the calcitic protrusion are rounded and differ from a typical finger-joint system. Hence, as the reviewer suggests, we corrected the phrase “finger-joint” into “knob-like protrusions” in the revised version of the manuscript.

Reviewer 2

General comments:

This work uses mechanical testing and finite element modelling to illustrate the roles of different structural features in one echinoid species' mineralized test (with regard to flexural rigidity of skeleton material, segments interlocking and collagenous sutures) to argue the mechanical importance of the test's tessellated design, for the animal's ecology, but also for bioinspired design. The topic and system are relevant and timely for current interests in structured biomaterials research and sea urchins have proved a useful model system for bioinspired architecture lately. However, there are some issues with the design of the experiments and manuscript that require restructuring and/or deeper discussion:

Thank you for your positive comments and constructive criticism. We sincerely hope that including all the requested modifications and explanations we will be able to increase exhaustiveness and interestingness of our paper.

1. Language: A comparatively minor point – the work suffers from a variety of grammatical issues, particularly subject-verb agreements (e.g. plural noun, singular verb – e.g. [page 2, line 50], [page 3, line 29]) and misspellings (even of the study species' name), but also many sentences with convoluted phrasing that is difficult to follow (I point out instances below). This is likely the result of translation, but requires more stringent polishing.

We have proofread the entire paper to check grammatical issues.

*2. Scope: There is an odd mismatch between the purported scope of the paper and the actual content. Whereas the intro and even the title suggest the work will cover a much broader range of topics fundamental to how echinoid shell structures are designed in general, the core of the work is really a focused study of some specific aspects of plate-plate actions/features in a single species. It then reads as if the authors were pushing to make the work sound 'bigger' than it actually is, which only serves to dilute some nice messages here. I would find the study much more convincing if much of the extraneous 'fat' were trimmed away to center on the core research questions relating to *P. lividus* (e.g. the very broad cook's tour of ideas in the introduction could be condensed considerably and moved to an 'implications' section in the conclusions). Along these lines, it needs to be clearer why we should be invested in looking at this species, given all the work that has apparently already been done on urchins. It wasn't until [page 5, lines 3-5] that I got a sense even of the actual scope of the work and the motivation (although I am still unclear what the mentioned 'hypothesis' is).*

We thank the reviewer for giving us the opportunity to better specify the main goals of our research. Our paper has the main goal of highlighting the beneficial effects that flexible sutures have on the mechanical behaviour of shell structures, either natural or artificial. This idea has risen from the test of *Paracentrotus lividus*, but with some exceptions, can be used to explain the mechanical behaviour

of many shell structures that present similar features, either natural or artificial. It even finds application in many fields of structural engineering and, as we have shown, in the design of shell roofs. The actual scope of the paper and its field of application is now clearly declared in the title, which has been modified to “Flexible sutures reduce bending moments in shells: from the echinoid test to tessellated shell structures.” The Abstract and the Introduction have been updated accordingly so that the actual scope of our paper is clearly stated.

Regarding our particular reference to this species, there are many studies on sea urchin tests reported in the literature. However, as discussed in detail in **comment 1 to reviewer 1**, presently no studies have investigated the effect of flexible structure on the mechanical behaviour of the entire test in regular sea urchin and in particular on their role in reducing bending moments. Previous literature indagated the mechanics of segmented shell structures by focusing on singular aspects such as the strength of sutures [Ellers et al., 1998], the role of the trivalent vertex [Wester, 1984; 2002] or the behaviour of the particular clypeasteroid shell considered to behave as a monolithic structure (Grun and Nebelsick, 2018). The hypothesis “the structural organization of the test, which is a composition of multiple rigid plates joined by flexible sutures, can significantly reduce bending moments so that the test shell is capable to resist external loads mainly by membrane forces, without compromising the global deformability and stability of the shell,” (**lines 69-72**) instead, is more general and identifies a reason for the increase of strength, documented by some authors (e.g. Ellers et al., 1998), and the restriction of relative rigid displacements between plates, described by others (e.g. Wester, 1984; 2002). Although directly applicable to interpret the mechanics of *P. lividus* test, this behaviour can be easily generalized to any tessellated shell, in which sutures are at least one order of magnitude more flexible than plates. The mitigation of bending moments caused by flexible sutures is not a specific property of the *P. lividus* test but, with due exceptions, can be extended to a class of shell structures, all exhibiting the same features. These features are the following: curvature of the shell mid-surface; small thickness when compared to the shell curvature; shell subdivision into rigid plates and deformable sutures (or joints); curved geometry of plate edges; sutures (or joints) organized according to the trivalent vertex principle. These characteristics are common both to the analysed echinoid test shell, and to some representative examples of shell roofs. These structures are very different from the test of *P. lividus*, not only for the scale of the structure, but also for their shape and for the magnitude, distribution, and origin of loads the structure is demanded to bear. Although these differences, their mechanical behaviour exhibits the same pattern: mitigation of bending moments due to the presence of flexible sutures.

We are certainly aware that our experiments are limited to the test of *P. lividus* and, for this reason, it is imprudent to generalize these observations to other species. However, it is quite common in structural mechanics, to derive a general result from single case studies if the generalization is obtained from a well-founded theory (theory of elasticity) and is supported by numerical results relying on this theory (FE analyses). Hence, trying to further motivate our generalization, we reported a new set of FE analyses in the revised version of the manuscript (section 3.4, better documented in the supplementary material), where we have parametrically varied the geometry of the test, the relative stiffness between sutures and plates and between sutures in different regions of the test (**lines 384-400**). As a result, it is shown that the global effect on the distribution of bending moments within the test is only marginally affected by these variations.

3. Experiments: The experiments themselves are also admittedly quite limited: as far as I can tell, the mechanical tests involve only one or two samples for each condition and the biological FEA models include one monolithic model and one segmented model?

Thank you for your observation. The reviewer is correct. Our previous set of experiments was rather limited and were not sufficient to comprehensively characterize the mechanical behaviour of sutures. For this reason, we decided to repeat all experiments and extend their number. This has been also motivated by a flaw that we found in previous experiments. New experiments have been double checked and are documented in section 3.2 of the revised version of the paper. In particular, we performed three-point bending tests on a total number of 10 samples of interambulacral plates and 10 plate-plate pairs, for a total of 20 different experiments (**lines 240-243**). These results are sufficient for the scopes of our paper and confirm the high flexibility of sutures.

Regarding the number of FE models, the elaboration of our experiments allowed us to carry out numerical analyses on both a monolithic and a segmented model of the echinoid test loaded by 18 different load conditions for both the monolithic and the segmented models (**lines 349-354**). In particular, we considered the following loading conditions:

- 1) Uniform normal pressure of -8,2 Pa applied to the entire shell. It used to account for the different internal celomic / external environmental pressure.
- 2) Lateral force of 10N uniformly applied to the entire shell.
- 3) Normal pressures having resultant of 10N applied over 16 different regions having different area and position with respect to ambulacral and interambulacral plates.

The results of these analyses have been included within the supplementary material attached to the revised version of our manuscript. In particular, conditions 1 and 2 produce mainly membrane forces and very limited bending moments both on the monolithic and the segmented models. This behaviour is common to almost every doubly curved shell, at any scale, and is well known in structural engineering. This behaviour can be extended to tessellated shells. Localized loads, instead, are very dangerous for thin shells and are responsible for punching failure. Actually, the results regarding the 16 loading conditions described at point 3 show that these loads produce higher values of bending and out-of-plane shear in the vicinity of the loaded area. As proved by our analyses, the presence of flexible sutures rapidly mitigates these effects in points surrounding the loaded areas. This behaviour is similar in all considered cases and for this reason, we decided to include just one representative result in the previous version of the manuscript (Figures 8 and 9). However, in response to the reasonable request of this reviewer, we included all additional results in the supplementary material attached to the revised version of the manuscript. In addition, we considered and included additional new results relevant to varied test geometry and bending stiffness of sutures and plates (**lines 379-400**).

On P4 20 and 49, the authors mentioned shell structures are optimized for both bending and out-of-plane shear, however, there is no support given for this statement.

On P4 20 and 49, we wrote “bending and out-of-plane shear actions are associated to an uneven employment of material” and “Accordingly, an optimized shell structure works mainly under membrane actions and reduced bending and out-of-plane shear.” Beside the nomenclature “actions” that has been corrected to “forces” in the revised version of the manuscript, we did not state that “shell structures are optimized for both bending and out-of-plane shear” as the reviewer refers. Additionally, as it can be easily inferred from the formulas (2) in section 3: bending moments produce uneven distribution of normal stresses, i.e., normal stresses are a function of z , while the membrane forces stress the material uniformly, no dependence on z is present in the terms associated to membrane forces. A direct consequence of this well-known behaviour [Ventsel & Krauthammer, 2001] is that the employment of material is optimized (all material fibers pertaining to a chord are stressed in the same way so as to avoid that any material fiber is under stressed with respect to its strength) only if

the shell works under pure membrane forces. Clearly, this is not a necessary condition since the distribution of membrane forces can still be uneven and some parts of the shell structure can be more or less stressed than others, also depending on the eventual non-uniformity of the shell thickness. Of course, the matter of structural optimization is very broad and in need of further research [Adriaenssens et al. 2014], which is beyond the scope of our research. In conclusion, in order to avoid any further misinterpretation of our statements, we rephrased the sentences mentioned by the reviewer (lines 127-132).

To explore this more, I would recommend adding shear stiffness tests and other load cases to the FE analysis.

Although the name might be misleading, three-point-bending tests produce both bending and out-of-plane shear within the specimen. The value of shear is equal to half the applied load on both sides of the beam, with opposite sign. In addition, we observed no relative sliding between the two halves of the plate-plate system, neither we experienced a shear failure, either at sutures or within the plate (lines 261-264). The failure mode of plates, which is governed by bending, is certainly due to the reduced thickness of the shell, while relative sliding and shear failure at sutures is probably prevented by the diagonal collagen ligaments and by the high interlocking effect produced by the knob-like trabecular protrusions.

We also notice that a shear stiffness test is very difficult to perform at the limited scale of the test shell: it requires a working span that has approximatively the same size of the shell thickness (about 1mm). Hence, apart being difficult and expansive, this test is inessential to the scopes of our research.

Although the authors claim the morphology is ‘deeply investigated’, this is presented only in quite small images in Figure 2, with most data relating to architectures that can be viewed from the surface only. Given that the mechanical behavior of the sutures is believed to be a function of both surface and ‘through-thickness’ interactions of the plates (e.g. collisions of the sutural ‘fingers’), the presented analyses are insufficient. Images showing the finger-collisions (e.g. from microCT scans of plates loaded in in-plane shear and bending) and/or tighter relations of the morphologies to the shown loading curves would be much more convincing. Relating to the previous point: a smaller, tighter paper centered on the more focused questions of plate-plate interactions in this species, then those finescale data reinforced by the biological and bioinspired FEA data, would be an entirely interesting and useful contribution – I would recommend the authors reframe the work in this regard.

As suggested by the reviewer we clearly stated the main goals of our work. As stated above, it is focused on the global effect produced by the plate-plate interaction. Future works will be focused both on the suture morphology, micromechanics of plates and plate-plate interaction. Additionally, in order to address this reviewer’s request new images that illustrate the morphology of the analysed interradial sutures have been added in Figures 1, 3 and 4.

4. Loading environment: I recognize the work that goes into establishing a functional FEA model, but the load cases investigated seem to be centered largely on bending, despite the service paid to other loading regimes in Fig. 1.

As mentioned earlier (see, e.g., our response to the first part of Comment 3), 18 different loading conditions have been investigated and the most interesting ones correspond exactly to those that

produce high values of bending moments since this is the main goal of our research. Actually, reported results show the beneficial effect of flexible sutures since they reduce bending moments. In the revised version of the manuscript this goal is very clearly stated and all considered load cases are reported (**lines 349-354**). Different load cases would not be useful to verify the role of flexible sutures in reduce bending moments since:

- (1) uniformly distributed pressure already produces membrane forces, this is shown by the results reported in the new supplementary material (load cases P and L).
- (2) loads applied parallel to the shell mid-surface are already acting in the direction of membrane forces and produce small bending and out of plane shear.

Accordingly, the only way to put under severe verification our hypothesis, that in short is “flexible sutures reduce bending”, is that of loading the shell in such a way to produce high value of bending and verify that these components are mitigated by the presence of flexible sutures.

However, in order to fulfil the reviewer request, we report the results obtained from all considered loading cases (even the less interesting ones) in the supplementary material attached to the revised version of the manuscript.

Also, if ‘skeletal material tends to increase in density in different regions’ (pg. 5) wouldn’t this also be important to the mechanical behavior? And yet, in physical tests (stiffness of single plate), structure is ‘imagined as homogenous and isotropic’ P7 18. Incidentally, the equations discussed at the start of section 2 (‘Investigation of the echinoid test’) are interesting, but not really supported or explored here in any way, neither by citations nor tests to validate them. It’s therefore unclear if that section is meant to be a result or is some consolidation of known theories? Currently, it feels quite ‘tacked on’.

The microstructure variability and the mechanical behaviour of the skeletal plate is the focus of our next work. Material density is diversified in two manners: a) the density and organization of microstructure is differentiated within each plate and b) the presence of pores in ambulacral plates and the small variation of shell thickness modifies the stiffness of some regions of the shell. In the first case the inhomogeneities produce only local effects that can be studied by employing a model of just a few plates. These effects are however negligible at the global scale, which is the focus of the present work. Regarding the second kind of inhomogeneities, these can produce a variation of the global behaviour of the shell. In order to fulfil reviewer’s request, we investigated their effects by varying the stiffness of ambulacral plates, that has been reduced by a factor of five (**lines 394-397**). These results show that even in this case bending moments are sensibly reduced by the presence of flexible sutures.

Regarding the equations reported at the beginning of section 2, they serve as an introduction and motivation to the problem we are going to face in the remaining part of the paper. The definition of internal forces in shell structures is needed to present the quantities we are going to compute in our FE analyses. The distribution of Cauchy stresses associated to the several components of internal forces motivates our focus on the value of bending moments, while neglecting all other components. Also, they are functional to understanding the basic concepts of shell optimization, a matter that is not familiar to every potential reader of the paper. Actually, being ours multidisciplinary research, we expect not all potential readers are experts of both biomechanics, structural mechanics and structural optimization. For this reason, we believe it is useful to recall some basic concepts needed to fully understand and interpret our results.

Specific comments: listed by [page number of reviewer copy, line number]

[2 42] *Here immediately, but also throughout the intro, the context isn't clearly framed. A good example of this is that the study system isn't shown until Fig. 2 and even then the anatomical arrangements aren't clear and the images are small. I'd recommend a figure more like Fig. 1 of this paper (<https://www.sciencedirect.com/science/article/abs/pii/S1047847714002792>) as a first image to build context.*

The title, the abstract and the introduction have been rewritten in order to make clear the goal of presented research, e.g., see also our response to a previous observation by the same reviewer. Additionally, to accomplish reviewer request about the anatomical arrangement of the *P. lividus* test, a new image illustrating the morphology of the test has been added in the revised version of the paper, see, i.e., Figure 1.

[2 48] *I found this sentence hard to follow at first, perhaps because I had never heard of 'membrane actions' before (a concept mentioned over and over). Please define.*

We corrected the use we made of the word "actions", which has been substituted either by "force(s)" or "moment(s)" in the revised version of the manuscript. Accordingly, "membrane actions", "bending actions" and "shear actions" have been corrected to "membrane forces", "bending moments" and "shear forces".

The definition of membrane forces and membrane stresses are paramount concepts of shell mechanics and are defined in the first chapters of the books by Calladine (1989) and by Ventsel & Krauthammer (2001) mentioned in the paper. The entire first part of section 3 has exactly the goal of making the paper self-contained and the reader, that potentially is not necessarily a structural engineer, aware of the internal force components in shell structures and how they influence the structural optimization (see also the previous observation by this same reviewer, where the definition demanded here has been considered superfluous).

As already recalled at the beginning of section 3 (**lines 112-113**), membrane forces are defined as a component of the resultants of normal and tangential Cauchy stresses along the thickness of an infinitesimal region of the shell. Considering the orthonormal axes x and y parallel to this plane, the Cauchy stresses that contribute to the membrane forces are the normal stresses σ_x and σ_y and in-plane tangential stress τ_{xy} . Their resultants are computed along the thickness of an infinitesimal element $dx dy$ of the shell and form the membrane force tensor, of components N_x , N_y and N_{xy} . Here subscript directly refer to the corresponding Cauchy stress components.

[2 58-59] *Throughout the work, statements like this need citations for support.*

The structural performance of corrugates shells is well known in structural engineering. It is a widely used engineering design technique to obtain stronger and stiffer shells with reduced employment of material [Calladine 1989, Ch. 10]. One of the earliest and famous application of this concept to large scale structures is the roof of the Aircraft hangar at Orly Airport, Paris by Eugene Freyssinet (1921), or the more recent roof of the CNIT in Paris by Nicolas Esquillan (1958).

This same concept is largely employed in more common shell structures, at smaller scale, such as corrugated pipes and corrugated sheets

A theoretical justification of how corrugation reduces bending moments in shell structures can be derived by employing the solution for cylindrical shells given in Ventsel & Krauthammer 2001, but goes far beyond the scope of our manuscript.

Clearly, this same approach can be used to show that corrugation in seashells has exactly the same role as reported by Wainwright et al, 1976 pg. 262. This reference has been added in the revised version of the manuscript (**line 33**).

[3 4] Morphology and scale are hard to picture (see comment [2 42])

As mentioned in our answer to comment [2 42], a new figure (figure 1) has been included in the revised version of the paper in order to show the morphology and scale of the echinoid test.

[3 10] Biomimetic >> Biomimetics

Corrected

[3 22] Is deformability an important factor here?

Yes, deformability of the shell is increased by the tessellation. The role of deformability, however, is a scale depending issue [Charpentier and Adriaenssens, Effect of Gravity on the Scale of Compliant Shells. Holmes, Elasticity and Stability of Shape Changing Structures]. In particular, for small- or medium- scale shells this is not an issue, since double curvature of the shell allows to obtain very stiff structures and excess of deformability is rarely a problem. Additionally, structures subjected to

dynamic loadings such as ground acceleration, deformability combined with hysteresis of materials is helpful to reduce inertia forces and dissipate energy [Chopra, Dynamics of structures]. For very large shell structures, instead, deformability can be a fundamental factor for their design.

In order to show the effect of flexible sutures on the shell deformability, we included additional results showing the distribution of displacements for both monolithic and tessellated models. These results are reported in the supplementary material attached to the revised version of the paper.

[3 27] easier to read, if you put ‘neglecting’ after ‘while’ and cut ‘are neglected’. But at the end of this paragraph it would be useful to clarify WHY neglecting the MCT behavior is useful (e.g. since tissues like that are hard to mimic in engineering)

The sentence has been corrected. The problem of mimicking MTC regarded the lack of experimental data. This issue has been solved in the revised version of the manuscript by repeating all experiments either on samples immersed in filtered seawater and on samples immersed in 100 mM K⁺. The results of our experiments show an effect of K⁺ in modifying the bending stiffness of sutures [**see the answer 2 to reviewer 1**].

[4 14-15] Hard to follow – you’re trying to outline an isostress scenario? In general, I found the terminology and goals of this section difficult to follow (see General Comment #4 above)

Here we refer to a uniform distribution of Cauchy stress values along the thickness of the shell. Clearly, several regions of the shell can still be subjected to unequal stress values. This sentence has been rewritten to improve readability. (**lines 127-132**)

[4 Fig.1] (b) Check the arrow directions (V_x, V_y) for out-of-plane shear components

Thank you for spotting this. These arrows have been corrected (Figure 2).

[4 56-7] Words missing, unclear

The sentence has been rewritten in order to improve readability (**lines 141-142**).

[4 58-60] Again, statements like these are baseless assumptions without citations (see comment [2 58-59])

The sentence has been modified to increase clarity and a reference has been added as required by the reviewer (**line 144**).

[5 1] Words missing, unclear

The sentence has been corrected (**line 145**).

[5 7-8] Don't understand the meaning here.

The sentence has been corrected (**lines 72-74**).

[5 12-5] As stated above, the 'hypothesis' is quite unclear, but it seems the goal is to demonstrate how 'collagenous sutures' help to distribute loads by 'membrane actions' – how can that truly be tested though if the true mutable nature of the collagenous tissue isn't captured/modeled?

As stated before, the effect of MTC has been investigated in the revised version of the manuscript. See, e.g., our response to observation [3 27] by the same reviewer and the answer 2 to reviewer 1.

Also what does it mean to reduce 'bending actions'?

Regarding the meaning of “reduce bending moments,” this is the central message of the entire manuscript and the entire manuscript is focalized on supporting this hypothesis by numerical results, shown in figures 9, 10 and 11 and in supplementary material. Its implication on the optimization of the shell structure is clear from equations (2) and from the relevant explanatory text (**lines 124-134**). We believe no further explanation is required.

[5 32-4] Hard to picture without better anatomical context

The ball-and-socket joints are not essential for the scopes of our research. A description on how spines are attached to the test is only included in our paper for completeness. However, in order to refer the interested reader to a proper description of these joints we added a reference to another research in the revised version of the paper (**line 179**).

[6 -10] If much of this is already known from other studies, what is the justification for exploring it more deeply here?

Not everything needed for our study is already present in existing literature. The first (yet not the most important) merit of our research has been that of joining results consolidated in different fields such as the increase of strength observed by Ellers on echinoid tests, correctly attributed to the presence of collagen ligaments, and the inhibition of rigid mechanisms described by Wester for (artificial) structures composed of plates and cylindrical hinges. Notice that even the word “plates” has two different meanings for these two researchers. Ellers refers to the biological definition, i.e., the skeletal components of the echinoid test and, in this context, plates have double curvature have both membrane and bending rigidity; Wester, instead, being concerned with the kinematic of plate-hinges systems, refers to plates with the meaning of flat bidimensional elements that represent a rigid constraint for displacements in their plane.

Additionally, our manuscript contains several original aspects not investigated in the previous literature (at least according to our knowledge). These are:

Mechanical characterization of the bending behaviour of sutures. This has been done for the first time in our research by means of three-point bending tests on pair of plates joined by sutures. Sutures have been mechanically characterized by elaborating the experimental curves to obtain stiffness-rotation curves, which are mathematically described as a function of a few, well defined, mechanical parameters. This elaboration is described in section 3.2 of the revised version of the manuscript.

The effect of curvature of plates and curvature of edges on avoiding the relative rigid rotation between rigid plates of the *P. lividus* test has been described for the first time in our manuscript. This is the result of a very simple application of the Mozzi-Chasles's theorem to a system composed of two rigid plates joined by a curved cylindrical hinge. See also our answer to observation [6 14-5], hereafter. This is described in section 3.1 of the revised version of the manuscript and graphically shown in Figure 5.

Finally, and most importantly, the main goal of our manuscript is, as the new title clearly states, to show that flexible sutures reduce bending moments in tessellated shell structures. To our knowledge this has never been stated before and the entire manuscript is centered on this message. This behaviour, observed by us on the test of *P. lividus*, can be easily generalized to other shell structures, both natural or artificial, composed of rigid plates and flexible joints, provided that some specific structural requirements (synthetically listed in table 1) are fulfilled. To show that these requirements are sufficient to reproduce the same mechanical behaviour, observed on the *P. lividus* test, on shells of completely different scale, material and subjected to completely different loading conditions, we wrote the entire section 4 of our manuscript. This has never been done in previous research. Previous mimicking of echinoid test, done for instance by Knippers, is purely formal.

To clarifying what is missing in previous literature we added a few sentences at the end of the introduction of the revised version of the manuscript (**lines 57-58**). Additionally, to highlight our achievements, we included their short description in the conclusions of the revised version of the manuscript (**lines 488-524**).

[6 14-5] Why not reference figure 3 here? Also, how are the actions in that figure determined? Simply intuitively or has this been explored more methodically?

Reference to Fig 3A (Figure 5A in the revised version of the manuscript) has been added. This figure represents schematically the result of a very simple rigid kinematic analysis of the plate-plate relative motion. Actually, due to the curved geometry of sutures the axis of relative rotation is not unique hence relative motion between plates is prevented due to the Mozzi-Chasles's theorem. To eliminate any doubt about the method employed here, we explicitly mention that curved sutures avoid the existence of skew axis (also known as Mozzi axis) for the relative motion of plates in the revised version of the manuscript (**lines 230-232**).

[6 36-38] 'avoid' is an odd word choice here and sounds too personified. Do you mean perhaps 'restrict' or 'limit'? Also, would the nested curved plates in Fig. 3C/D even be able to rotate at all or would they tend to jam each others' movement?

We modified "avoid" to "restrict". Provided that the suture is not damaged, rigid motion is completely restricted. However, deformation of all parts is always possible.

[7 3-5] Nice experimental idea, hard to visualize though, I didn't understand it until later – what I really would have loved to see was a more specific demonstration of how microarchitecture related to the loading behaviors shown

Analyses on the role of microstructure on the distribution of stresses within plates is currently under investigation by our research group. According to some preliminary results obtained by very refined

FE analyses, the microstructure influences the local distribution of stresses within the single plates and has negligible effect on the global behaviour of the test. These aspects, although very interesting in a wider context, go far beyond the goals of this manuscript, which are focalized on the role of sutures in reducing bending moments in tessellated shells.

[7 11] ‘immersed’ >> do you mean ‘immersed’? (It’s spelled differently in English from Latin)

This typo has been corrected in the revised version of the manuscript.

[7 22-27] Already reported in previous paragraph; combine and condense

This sentence has been rewritten.

[Fig. 4] Is the blue line reaching yield?

In the revised version of the manuscript Figure 4 has been substituted by Figures 6 and 7, respectively referring to plates and plate-plate pairs. However, the blue line contained in the old Figure 4 is now corresponding to one of the 8 lines of Figure 6A. The last point of all these lines are corresponding to a brittle failure of the tested plate. This failure mechanism is clearly described in the revised version of the manuscript (**lines 260-261**).

[Fig. 5] Are these two separate experiments?

In the revised version of the manuscript Figure 5 has been substituted by Figure 7C and 7D. These curves do not correspond to separate experiments, but they are an elaboration of the experiments made on plate-plate pairs. The original experimental output is reported in Figure 7A (corresponding to the red curve of the old Figure 5). This was already described in the original version of the manuscript and this description has been reported in the revised paper also, see the text before formulas (3) and (4): “For the interpretation of the experimental results regarding the plate-plate pairs [...]” (**lines 273-292**).

[8 21-23] So are you saying the mechanical behavior would be roughly the same without the collagen? Doesn’t that go against your hypothesis? Could you dissolve the collagen and try that?

Not at all, we believe collagen has a significant role on the behaviour of sutures. Actually this was already proven by Ellers et al. (1998). However, mentioned description is misleading and requires a broader research and description. Hence, it is far from the main goal of this manuscript, and we decided to remove it from the revised version since it will be the object of our future research.

[Fig. 6C-E] Is this both the in-plane and through-thickness morphology?

Yes, this is both in-plane and out of plane morphology, but shown mechanism is in plane relative rotation, which produces out of plane displacements. As stated in our previous answer, this figure, together with the relevant description, has been removed from the revised version of the paper.

[9 2] calibre >> caliper

Corrected.

[9 10-11] Unclear, reword

This sentence has been rewritten.

[9 19] How? Wasn't that behavior non-linear?

The nonlinear behaviour of sutures is expressed by the function $K_t(\phi)$, see, i.e. formula (4). If one substitutes this expression into that of E_{sut} given by formula (5), one obtains that E_{sut} becomes a non-linear function of ϕ .

[9, 22] beding >> bending

Corrected

[9 32] Reword, make simpler – I assume you're saying you set both joint and plate moduli to the plate modulus value? It's not really the stiffness of a 'single plate' but of the plate material, yes?

Yes, the sentence has been rewritten

[9 41] Words missing, unclear.

The sentence has been rewritten.

[Fig 7] – what is the unit of FE analysis scale (strain /stress)? I'm confused how the test is loaded here – I don't see how you can end up with such localized stress(?) in D if the test is monolithic, unless that rounded rectangle shape is the shape of the loading zone? Is B really derived from A? I can't see plate shapes at all in the latter. Also, make C(a) and (b) bigger, hard to see.

We added units to figure 7 (now figure 9) and we described with more details the load conditions (**lines 349-358**). An image of loaded regions is also reported in the supplementary material attached to the revised version of the paper.

When loads are localized on a small portion of the shell, higher values of bending moments are computed in the vicinity of the loaded region. Similar results are obtained for all loading conditions corresponding to a pressure applied to a limited region of the monolithic shell. This localized effect is well known in engineering and causes punching failure both in monolithic and tessellated shells and we sincerely don't see why this reviewer is sceptic about these results.

Regarding the similarity between the surveyed and the analysed models, figures 7A and 7B (now 8A and B) were generated from different programs. In particular, the photogrammetric survey (Fig.8A)

is shown in a perspective view and lighten from above, while the parametrized geometry is shown in axonometric view and lighten from the left-upper corner. The updated version of figure 8A is correctly shown in axonometric view and now the similarity with fig 7B is much clearer although lighting is still different. However, notice that the aboral portion of the surveyed specimen was damaged and the aboral opening seems larger. This part of the shell was reconstructed in the parametrized geometry.

In order to remove any doubt, we include below a superposition between these two figures

Unfortunately, the geometry of single plates and sutures is barely visible from the photogrammetry since these superficial features are smaller than the photogrammetric survey accuracy (**lines 320-323**). For this reason, the size and location of sutures has been surveyed manually and, for this reason, their geometry was simplified.

Additionally, thanks to the employment of a parametric geometric model of the shell test, we were able to modify the shape by controlling a few parameters. This allowed us to perform additional analyses on a varied geometry of the test, showing how the capability of sutures to reduce bending moment is not related to the actual shape of the shell and thus it can be generalized to shells of different shapes such as those of different species. These analyses are mentioned in the revised version of the paper (**lines 384-386**) and documented in more detail within the supplementary material.

[10 36-7] How is this shown specifically?

As shown by our FE analyses, bending moments are reduced by the presence of the flexible sutures, while the global stability is preserved by avoiding rigid mechanism thanks to the trivalent vertex principle and the curved geometry of sutures. These aspects are widely discussed within the entire sections 3.1 and 3.4 of the revised paper and further recalled in the conclusions (section 5). Hence, we believe they do not need any further explanation.

[10 39-45] How are these known? Just assumptions or summaries of others' data?

Some of these requirements are discussed in section 3 and a reference to the relevant sections of our manuscript have been added (**lines 421-427**). For the other statements, we included proper bibliographic references.

[11 20-24] Interesting idea, but how are the model constraints arrived on (e.g. element size/shape)? The form-finding method described is unclear to me. Also, I've never seen funicular used this way, reword.

Element size has not been optimized since the main goal of this example is to show that what has been observed on the *P. lividus* test can be observed on completely different structures. Hence the element size has not been set according to any specific requirement. This aspect is however interesting and could be certainly investigated in future researches.

Regarding a detailed description of the form finding method, we believe this goes far beyond the scope of our manuscript. However, the reviewer can refer to [Adriaenssens et al. Shell structures for architecture: form finding and optimization] or to some methods developed by the first and last authors [Marmo et al. On the form of the Musmeci's bridge over the Basento river, Marmo et al., A novel approach to the form finding of shells based on membrane theory, IASS20. Argento et al. Shells' shape optimization based on R-Funicularity, IASS20/21. Marmo & Rosati, Form finding of compressed shells by the Thrust Membrane Analysis, IASS 2018].

In the same references, in all bibliography cited therein and in a very vast set of scientific literature on the design and optimization of shell structures, the word “funicular” is used exactly in this same way. It originates from the latin word “funis”, synonymus of “catena”. The notion of funicular structural forms has been used in design of shells since ancient times. Actually, the design methods based on this concept was hidden in a famous anagram by Hooke (“Ut pendet continuum flexile, sic stabit contiguum rigidum inversum”) [Hooke, A Description of Helioscopes, and Some Other Instruments." Phil. Trans: 1675] and explicitly recalled by Gregory in 1695 (“Catena in piano verticali, sed situ inversu, figuram servat nec decedit, adeoque arcum seu fornicem facit tenuissimum”) [Gregory, Catenaria, Philosophical Transactions, 1695]. See, e.g., Block et al. [As hangs the flexible line: Equilibrium of masonry arches. Nexus Network Journal (2006)] to find an exhaustive explanation.

[11 30] I don't see these effects at the centers of plates, please indicate.

As the colour legend shows, higher values are associated to red shades. These are clearly visible within figure 8 (figure 10 in the revised version of the paper). This has been further clarified in the revised version of the manuscript (**lines 460-461**). See also our answer to observation [12 11] below.

[12 8] meaning 'typical' for manmade architectures with all elements similar?

The sentence has been rephrased.

[12 11] Are peak values really the best for evaluating the performance? That would be heavily influenced by outliers. What about the top X% or some aspect of stress density (stress/volume of interest)? Here you say the reduction is minimal, but in the conclusions this reduction is played up.

We included additional diagrams showing the relationship between the value of the maximum principal bending moment computed at the quadrature points of the FE model and the number of quadrature points experiencing this value. These diagrams clearly show how flexible sutures (or hinges) reduce bending moments on the entire structure, not only their peak values. See figures 9C, 10 and 11 of the revised version of the manuscript and relative description.

[12 43-4] Not enough morphological variants were explored to really make this claim, in my opinion.

The parametrization of the test geometry allowed us to modify this shape and verify that bending moments are reduced by flexible sutures independently from the specific morphology of the test. These results are mentioned in the revised version of the paper (**lines 384-386**) and better documented within the supplementary material. Also, according to the results reported in sections 4.2.1 and 4.2.2, regarding shells of completely different scale, material, shape and subjected to completely different loading conditions, it is reasonable to affirm that this claim is well posed although its universality has not been proved. However, to avoid confusion, we rephrased our claim in the revised version of the manuscript.

Appendix B

Response to reviewers

We are grateful to both reviewers for their additional advises regarding our manuscript. Their suggestions have been carefully considered and addressed as follows:

Reviewer 1

Paragraph lines 180 to 204 is too long. I suggest subdivision at line 192 preceding "Considering...."

This paragraph has been subdivided as suggested by the reviewer

References 4 and 47 are repeated

Mentioned references have been corrected

Reviewer: 2

Comments to the Author(s)

General comments:

As previously, I greatly appreciate the topic of this research and the authors' efforts to draw on a relevant biological example for understanding the performance of tessellations in architecture. The authors have clearly taken many reviewers' critiques to heart, particularly evident in the expanded experiments and modified figures, the latter much more intuitive (although still requiring some tweaks, in my opinion; see below). Unfortunately, however, I still often found the prose overly dense and repetitive, which makes it very challenging to read, and therefore to access the nice science here. I'm not sure if this stems from disciplinary or language differences or something else, but I think having an outside editor would be invaluable: I would recommend the authors at least ask a colleague from another discipline to read the work and highlight text that is particularly unclear or confusing (I point out some areas I found difficult below). I believe some steps can be taken to make headway in this regard:

We thank the reviewer for the useful advice. The manuscript has been thoroughly revised in order to eliminate unnecessary repetitions and improve readability.

1. Often, phrasing feels as if it has been translated from another language, e.g.

135: "an optimized shell structure is capable to equilibrate"

299 and 401: "...are capable to allow"

333: "sutures permitted to obtain"

520: "which resulted to be sensibly modified"

...such errors would be easily caught by a proof-reader

We edited the manuscript to avoid such forms and, specifically, we corrected each of mentioned instances.

2. A minor point: I regularly tripped over the word 'test' – although it is the correct term for this shell, its other meaning is so common (especially in science) as to be a stumbling block. Perhaps 'echinoid test'?

We agree with the reviewer and substituted 'test' with 'echinoid test' throughout the manuscript.

3. Some information was noticeably repeated, e.g.

- *the background info from Ellers et al. and Philippi and Nachtigall (54-58, 151-155) – the discussion of this lit is very useful and appears to be in response to the other reviewer’s comments, but will be more powerful if condensed*
- *the extensive experimental summary, starting line 75, ‘gives away’ too many of the approaches and findings, immediately prior to the whole investigations section – this summary could be condensed by ~60% to provide a nice lead-in to the experimental section*
- *the Conclusion, while well-written, rehashes much of the discussion of results already found in individual “results” sections*

We condensed [151-155] to just one mention [line 158] in the revised version of the manuscript to avoid repetitions.

The summary starting at line 75 (line 78 in the revised manuscript) and the Conclusions are useful to describe what we have done and why. We edited these descriptions to improve readability but we avoided to condense them drastically. We also considered that many readers prefer reading the Introduction and the Conclusions before reading the entire manuscript.

4. Regardless of experimental details, a basic comprehension of the findings hinges on an understanding of some specific anatomical features and some engineering/mechanics concepts. With regard to the former, it would be useful to label all the features from the ‘Visual survey’ in Figure 1, I had trouble relating the text to the image. With regard to the latter, defining terms like membrane forces, bending moment, mid-surface, tessellation and shell structures early in the text would be very helpful (e.g. the shell definitions on 99-101 are needed earlier).

Fig. 1 has been improved in the revised version of the manuscript.

Mentioned definitions have been given very early, at lines 22-24 and 38-39, in the revised version of the manuscript.

5. The structure of the work is atypical – each ‘investigation’ section has its own methods, results and discussion (the latter rehashed in the conclusions, see #3 above). This can certainly work, but perhaps having a more standardized organization and more apt titling of sections would help the reader more quickly understand the format. For example, the ‘Visual survey’ section included an anatomical description (although the photogrammetric analysis came later, oddly), but then also a discussion of MCT and a structure-mechanics analysis. Perhaps it would help to organize the investigations by size scale (nicely done in Fig. 3), e.g. starting with general anatomy, then plate ultrastructures, then plate anatomical interactions, then plate and joint mechanics...

Though appreciating the suggestions of the referee for improving the clarity of the manuscript, we believe the current structure of the manuscript is the best choice for this specific topic, since it is very wide and ranges from bio-mechanics to civil engineering applications (conclusions are written to be read independently from the rest of the manuscript, see our answer to point 3). Anatomical description in the ‘Visual survey’ is organized according to the functional role of each part. For example, the joint mechanics is functional to understand why plates are not articulated at the global scale; hence, we believe that a description organized by size scale is not efficient in this respect. Incidentally, the Visual survey is a qualitative survey and fosters an holistic understanding of the global structural behaviour, while the photogrammetric survey, i.e. a quantitative geometrical survey, is just needed to accurately define the geometry of FE models. Hence it should not be surprising that the global perspective is presented first and the quantitative geometrical detail later.

6. I would argue that, in general, the prose needs to be tightened and made more accessible throughout. Again, an external reviewer/editor would be extremely useful here; I know that after a while, I certainly can become ‘too close’ to my writing and not see extraneous text! I could easily imagine the manuscript being shortened by ~30% and it would be far stronger for it.

As mentioned in our answer to the general comment by this reviewer, the manuscript has been edited in order to eliminate unnecessary repetitions and improve readability.

Specific comments: listed by [line number]

(PLEASE NOTE: up to line 150, I detail all comments I had, to give a sense of the scope of what I saw as regular critiques for the manuscript – many of these I imagine would be fixed through careful proof-reading by an external editor; after line 150, I highlight only comparatively larger issues)

[5] Limitless is not accurate & On the other hand ‘several’ is also used incorrectly (it should be used to mean >2, but not many) on lines 31 and 50

Both issues have been addressed in the revised version of the manuscript.

[18-20] Does an efficient structure necessarily effectively protect human activities?

Mentioned sentence has been rewritten in the revised version of the manuscript.

[22-23] Isn’t structural efficiency always related to the distribution of internal forces, not just in shelled structures?

Mentioned sentence has been rewritten to give it a broader meaning.

[32-36] Is this needed? Why not go right into tessellations?

We believe mentioning corrugation is needed here. Tessellation is not usually used in engineering, while corrugation is more familiar to the reader since it is a traditional approach to increase stiffness and strength of shell structures.

[48-49] What do you mean ‘expand’ – grow?

Expansion of plates is due to growth. This word is here used to describe the geometric transformation rather than the biological cause. Incidentally, the same word is used in reference 23 with the same meaning.

[50-54] Repetition of the strengthening action concept, even within these few lines. But the bit from 50-52, in my opinion, is one of the big keys that distinguishes your work (it just gets lost in all the other piles of information here, unfortunately)

We added a new sentence [54-56] to highlight the goals of our manuscript.

[74] Odd wording: “effect...on the value of bending moments” (if I get what you mean, could probably just remove “the value of”)

We removed “the value of” in the revised version of the manuscript.

[76] What do you mean “avoids rigid mechanisms”?

We added a sentence to explain the meaning of “rigid mechanisms”

[88] Odd wording: “after recalling the advantages of designing”

This sentence has been rephrased in the revised version of the manuscript.

[102-103] Showing these force directions on one of your figures would be useful

These forces (and their directions) are shown in figure 2, whose caption has been improved.

[137] “to make it funicular” – you use this phrasing several times, perhaps it is a normal phrasing in architecture or engineering? I have never heard it before and it reads oddly.

The definition of ‘funicular shell’ has been given in lines 139-140 of the revised manuscript.

[143-144] Careful, just because a morphology evolved, doesn’t mean it is the most optimized for weight (it could be optimized for something else, most likely for multiple, even competing factors).

Mentioned sentence has been rewritten to improve its meaning and address this issue.

[146] Support metabolic rate statement with citation

A reference has been added as suggested by the reviewer.

[150-151] A key point, but the logic leap to get here is not intuitive, maybe because the terms aren’t clearly defined. Yet, in the following sentence, these physical phenomena/structural behaviors are taken as fact.

Since the needed definitions are given earlier in the revised version of the manuscript, the meaning of this sentence and its logic should be now clearer.

[160-167] How do you know this treatment didn’t alter structure? Which plates were examined? (Relates to the question of local variation in morphology)

The use of NaOH to provide clean skeletons without altering their structure is reported as a standard protocol in many references, e.g., see:

Carnevali, M. D. C., Bonasoro, F., & Melone, G. (1991). Microstructure and mechanical design in the lantern ossicles of the regular sea-urchin *Paracentrotus lividus*. A scanning electron microscope study. *Italian Journal of Zoology*, 58(1), 1-42..

Martínez-Melo, A. (2019). Miocene echinoids from Palenque, Chiapas, Mexico. *Journal of South American Earth Sciences*, 95, 102258.

[180-186] Hard to follow descriptions – clear labels in figure would help

Figure 4 has been improved by adding labels.

[201-203] Reword, hard to follow (also 306, 328-331, 368-370)

Mentioned sentences have been rewritten.

[208-224] This is more like a typical ‘discussion’ – see comment #5 above

This is a description based on knowledge already available in the literature, but functional to our study.

[220] This and other discussions of the trivalent pattern in the text are hard to follow

A full description of the trivalent pattern is provided by Wester. His works have been cited several times in our manuscript and we believe that there is no need to further repeat descriptions reported therein. We only recall his results: trivalent vertices prevent rigid body mechanisms (i.e., articulation of plates) in CLOSED tessellated shells. The echinoid test is an OPEN tessellated shell, hence Wester’s result is not applicable as is. For this reason, we included the kinematic analysis of curved plates. This is clearly stated in our manuscript (lines 224-239)

[226] For the later joint loading tests, were plate-plate samples cut to avoid the jamming behavior you’re describing for curved tiles?

Plate- plate samples have a transversal width of about 1mm and at this scale the effect of plate curvature is negligible. We have better clarified this point in the text (lines 250-251).

[261-272] Key points about the behavior of tiles vs tile-joint composites, but this could be clearer – despite the discussion typically being wrapped in with the results in your paper format, there isn’t enough context here. Explain how the stress-strain curves illustrate what you’re describing in 261-264. How is the rotation demonstrated, is this shown in any figure? “It turns out...” is very casual wording. Why is the modulus written with a division sign? Is this modulus reasonable for the materials studied? There must be some precedent... etc.

As the reviewer pointed out, rotations are demonstrated only after the experimental curves have been elaborated. Accordingly, we modified this sentence (lines 268-268) to describe only the experimental curves, while their elaboration is described later (lines 301-311).

[297-302] The attribution of mechanics to structural aspects is made without clear support (e.g. is “damage of the suture” shown?). Also, the curves/trends in Fig. 6-7 are very difficult to see.

Damage is shown in curves of Fig. 8A and 8C: strength loss after the peak value denotes damage. There is no need to follow each single curve to catch this general behavior since the series of curves reported in Fig. 7-8 show clearly the general trend.

[325-328] Incorrect numbering (should be 2.4). How was the parameterized model built?

Numbering has been verified throughout the entire manuscript. We included required description regarding parameterization in lines 332-333.

[349-354] Nice range of experiments, but they feel a bit random – e.g., what relevance do the lateral forces and normal pressures have?

These loading conditions are not random. The resultant of applied lateral force is the half the vertical one employed by Philippi and Nachtigall (1996). The value of the normal pressure correspond to the observed celomic pressure reported in Ellers and Telford (1992). Both references have been mentioned in the revised version of the manuscript.

[383-384] Again, rigid-body mechanisms need to be defined

It has been defined earlier in the revised version of the manuscript. See previous response line 76

[400-407] Interesting points, but aren't all backed up with experimental data. E.g., I have yet to see demonstration of these rotations? Maybe I'm missing them in a figure? Also, I thought the material along the thickness wasn't homogeneous? And what evidence do you have to say the shell is an optimal compromise between mechanical support and growth?

Rotations have been experimentally observed and are determined quantitatively by elaborating the experimental curves of Fig. 8A by means of the mechanical model shown in Fig. 8B. This model accounts for a relative rotation between plates by means of the rotational spring at the midspan. It well interprets the curves of Fig. 8A and furnishes those reported in figure 8C and 8D. Here the rotation values are shown on the horizontal axes. Notice that if this rotation was absent, then we would obtain infinite values of K_s and K_t (they represent the secant and tangent stiffness of the rotational spring, hence infinite values of K_s and K_t mean no rotation). Clearly these figures show that we do not obtain these infinite values and our mechanical model well explains and describes our experiments. This procedure is fully detailed in section 3.2 of previous and current versions of the manuscript. However, in order to show our experimental setup and display rotations clearly visible after each test on plate-plate samples, we included a new figure (Fig. 6) showing photographs taken on one of our samples during the experiment.

[418] If this was in the equations in 2a-2f, this could be underlined more clearly for non-mathematicians...

The meaning of equations 2a-2f is described in section 3, just after the mentioned equations.

[424-427] Section numbering is wrong here as well and the wording of #5 is hard to follow.

Section numbering has been corrected throughout the entire manuscript. #5 has been rewritten.

[511] Again, I'd avoid dramatic words like "vast" & Maybe just "...generalized to a diversity of tessellated architectures"?

Mentioned sentence has been rewritten.

[Fig 3] This figure looks great, nice job – relating to the whole test would be useful, maybe with a little icon of the shell? What is 'ao'? Scale bars in the images would help

This figure has been improved by adding relations between all sub-figures and by adding a global view of the echinoid test. Also, the caption of this figure has been rewritten and the meaning of a_0 has been added.

[Fig 4] Nice looking images, but it's a bit hard to relate all the SEM images to the anatomies in A and to each other (e.g. it took me a moment to understand how C and D relate)

Relations between all sub-figures has been added to improve this figure.

[Fig 9] If the structure is modeled with homogeneous material properties, why are the stresses confined to one tile? Was the applied load exactly the size of the tile? What are the quadrature points?

Two detailed views of the loaded region (solid rectangle) and the plate boundaries (dashed lines - also deducible from the element mesh) have been added to Fig.9 (Fig. 10 in the new version of the manuscript). It is now clear that, as expected, the stresses are not influenced at all by the presence of plates in the monolithic model.